



# Combined organic and inorganic source apportionment on yearlong ToF-ACSM  dataset at a suburban station in Athens

Olga Zografou[1], Maria Gini[1], Manousos I. Manousakas[2], Gang Chen[2], Athina C. Kalogridis[1], Evangelia Diapouli[1], Athina Pappa[3], and Konstantinos Eleftheriadis[1]

5 [1]Environmental Radioacivity Laboratory, Institute of Nuclear and Radiological Sciences and Technology, Energy
Safety, National Centre of Scientific Research "Demokritos", Ag. Paraskevi, 15310, Greece
[2]Laboratory of Atmospheric Chemistry, Paul Scherrer Institute, CH-5232, Villigen PSI, Switzerland
[3]Laboratory of Inorganic and Analytical Chemistry, Department of Chemical Engineering, National Technical University of
Athens (NTUA), 9 Iroon Politechniou St., 15773 Athens, Greece

10 *Correspondence to*: Olga Zografou (o.zografou@ipta.demokritos.gr) and Konstantinos Eleftheriadis
(elefther@ipta.demokritos.gr)

**Abstract.** The current improvements in aerosol mass spectrometers in resolution and sensitivity, and the analytical tools for mass spectra deconvolution, have enabled the in depth analysis of organic aerosol (OA) properties. Although OA constitutes a major fraction of ambient aerosol, the overall aerosol properties are determined by the mixing characteristics of both organic and inorganic contents of ambient aerosol. In the present study, the mass spectra of both organic and inorganic aerosol were obtained by a time–of–flight aerosol mass spectrometer (ToF-ACSM) and further merged into one input matrix for Positive Matrix Factorization (PMF) analysis. The scope of this work was to assess the sources of organic aerosol and total non–refractory species in the suburbs of Athens, check their temporal variation and the interactions between organic and inorganic species, after reaching environmentally reasonable solutions for both matrices. The results revealed five factors in the case of the organic aerosol matrix. Three of them were primary OA factors: hydrocarbon–like (HOA), cooking related (COA) and biomass burning (BBOA), and the remaining two were secondary, less and more oxidized oxygenated organic aerosol (LO-OOA and MO-OOA, respectively). The relative contributions of these factors were HOA 15 %, COA 18 %, BBOA 9 %, MO-OOA 34 % and LO-OOA 24 % (yearly averaged). In the case of the combined aerosol matrix, two additional factors were identified that were mainly composed of ammonium sulfate (83.5 %) and ammonium nitrate (73 %). Moreover, the two secondary factors with both organics and inorganics were named as more (MOA) and less oxidized aerosols (LOA). The relative contributions on a yearly average of these factors were HOA 6 %, COA 9 %, BBOA 6%, Ammonium Nitrate 4 %, Ammonium Sulfate 28 %, MOA 23 % and LOA 24 %. The results showed a variation in secondary aerosols composition of organics and inorganics, mainly in less oxidized aerosol (LOA). This factor was composed primarily of organics during winter (80 %), while both organics and inorganics contributed equally to this factor in spring and summer; and in early autumn this factor presented more sulfate (70 %) than organics. This work presents a new methodology on ACSM data analysis, provides insights on the sources of the non–refractory species of ambient aerosols and using innovative tools for applying PMF (Rolling window) enables the study of the temporal variation of these sources and also the variability of their composition.



## 1 Introduction

The adverse effects of atmospheric aerosols on human health and the environment have been addressed by many studies (Ramanathan et al., 2009; Wilson and Suh., 1997; Pope et al., 2000; Jacobson et al., 2001). Particulate air pollution is one of the most important reasons for respiratory diseases (Dominici et al., 2006; Medina–Ramón et al., 2006). Apart from the negative consequences on human health, atmospheric aerosol may also be considered responsible for direct and indirect effects on climate. Ambient aerosols are mixtures of different chemical components that may cause both light absorption and

scattering (U.S.A EPA, 2012). For example, black carbon can absorb light at all wavelengths, brown carbon absorbs ultraviolet and visible radiation (Moosmüller et al., 2009), while organic aerosol, nitrate and sulfate particles are responsible mainly for light scattering (Cabada et al., 2004). Additionally, aerosol particles can act as cloud condensation nuclei (CCN) particles affecting clouds microstructure and lifetime (Rosenfeld et al., 2008). The overall effect of aerosols on climate and the aerosol–cloud interaction remain highly uncertain. Therefore, it has become essential to study ambient aerosol's physical

and chemical properties thoroughly.

Organic fraction usually comprises the greatest fraction of ambient aerosols (Kanakidou et al., 2005). Depending on their origin and formation process, they can be categorized either as primary (POA) or secondary (SOA) organic aerosols. They are considered as primary when they are directly emitted from a source, either anthropogenic or natural. Secondary are the organic aerosols that are generally formed through the oxidation of Volatile Organic Compounds (VOCs). VOCs quickly

evaporate after their emission and react with oxidants, such as hydroxyl radical ($OH^-$), ozone ($O_3$) and $NO_3^-$ radical, to form semi–volatile and low–volatility organic vapours (Robinson et al., 2007), which then condense onto pre-existing aerosol forming secondary organic aerosols (SOAs). SOAs are the dominant form of organic aerosols and can stay in the atmosphere long enough to undergo continuous oxidation and growth via coagulation and gas to particle condensation.

Inorganic species also comprise a significant fraction of atmospheric particulate matter. Sulfate is generally found in the

atmosphere as $(NH_4)_2SO_4$; ammonia reacts with ambient sulfur dioxide ($SO_2$) to form ammonium sulfate ($(NH_4)_2SO_4$). Nitrate is primarily found in the atmosphere as ammonium nitrate ($NH_4NO_3$). Ambient ammonium nitrate is formed through the oxidation of anthropogenic NOx emissions (NO and $NO_2$) to nitric acid ($HNO_3$), which eventually reacts with ammonia ($NH_3$). Ammonia is emitted into the atmosphere from different sources and processes, such as biomass burning, vehicles emissions, livestock emissions, the use of $NH_3$ based fertilizers and pesticides in agriculture etc. (Behera et al., 2013;

Schlesinger and Harley, 1992). Chloride containing particles are also released in the atmosphere during biomass combustion or are found in the form of $NH_4Cl$ (Lobert et al., 1999).

Over the years, mass spectrometry instruments have gained more reliability since their time resolution, sensitivity and selectiveness have improved, making them a powerful tool for chemical on–line and real–time characterization of ambient particulate matter. Time of Flight Aerosol Chemical Speciation Monitor (ToF-ACSM) is a descendant instrument of the

Aerosol Mass Spectrometer (AMS), which enables the quantification and chemical characterization of the non–refractory $PM_1$ species (species that rapidly vaporize at 600 ºC under vacuum conditions): organic, sulfate, nitrate, ammonium and



chloride by real–time measurements (Fröhlich et al., 2013). The application of different source apportionment (SA) techniques (e.g. PMF) on the derived mass spectra has enabled the in depth investigation of the sources and formation processes of organic aerosols (Ulbrich et al., 2009; Crippa et al., 2014; Zhang et al., 2019).

Previous studies on particulate matter source apportionment in Greece have mainly focused on inorganic datasets (Karanasiou et al., 2009; Argyropoulos et al., 2017; Diapouli et al., 2017; Manousakas et al., 2017; Manousakas et al., 2020; Manousakas et al., 2021), while only a few of them centred on measurements of the organic fraction measured by aerosol mass spectrometers (AMS/ACSM) (Stavroulas et al., 2019; Florou et al., 2017; Kostenidou et al., 2015). SA of organic aerosol is typically performed using the PMF algorithm. One of the latest advances in source apportionment modelling is the

rolling window technique (Parworth et al., 2015; Canonaco et al., 2021; Chen et al., 2021; Tobler et al., 2021) that is based on the modelling of a moving period of the initial dataset at each iteration. This technique has been found useful in order to examine the temporal variation of the identified factors and especially for the oxygenated organic aerosols, whose chemical footprint can vary in different seasons. A few studies have included in source apportionment schemes both the organic fraction and the inorganics from mass spectrometric instruments (Sun et al., 2012; McGuire et al., 2014; Hao et al., 2014;

Äijälä et al., 2019). All of these studies revealed that the inclusion of inorganics in SA studies improves both the solution obtained and the understanding of atmospheric processes and mixing between species.

Although source apportionment studies on organic aerosols for long periods have been prevailing in recent years covering a wide range of different sites, a long period of combined organic and inorganic source apportionment has not yet been published, leaving a gap in the comprehensive understanding of ambient aerosol sources, formation processes and mixing.

This study is the first one to present the results of two PMF analyses, one on the organic fraction and one on the combined organic and inorganic dataset of a ToF-ACSM, for a yearlong dataset enabling also the technique of the rolling window to examine the temporal variability and the varying composition of the combined factors. A comparison between the two solutions was performed and the mixing of organics and inorganics in different seasons was investigated. The validity of the retrieved factors was assessed based on the model residuals, the statistical uncertainty of each one of the retrieved solutions

and their correlation with specific external data.

## 2 Methods and Instrumentation

### 2.1 Measurement site

Measurements were performed at the Demokritos station (DEM), member of GAW and part of the ACTRIS and PANACEA infrastructures (37.995° N, 23.816° E), at 270 m above sea level (a.s.l.) (Eleftheriadis et al., 2021). The station is located

within the National Centre for Scientific Research (NCSR) "Demokritos" campus, a vegetated area at the foot of Mount Hymettus, about 8 km to the North east from Athens city centre (Fig. 1). The measurement site can be considered representative of the atmospheric aerosol in the suburbs of the Athens Metropolitan Area, since during the day it is exposed





to pollution transported from the urban area of Athens under most atmospheric conditions (western wind direction), whereas it is also occasionally influenced by incoming regional aerosol.

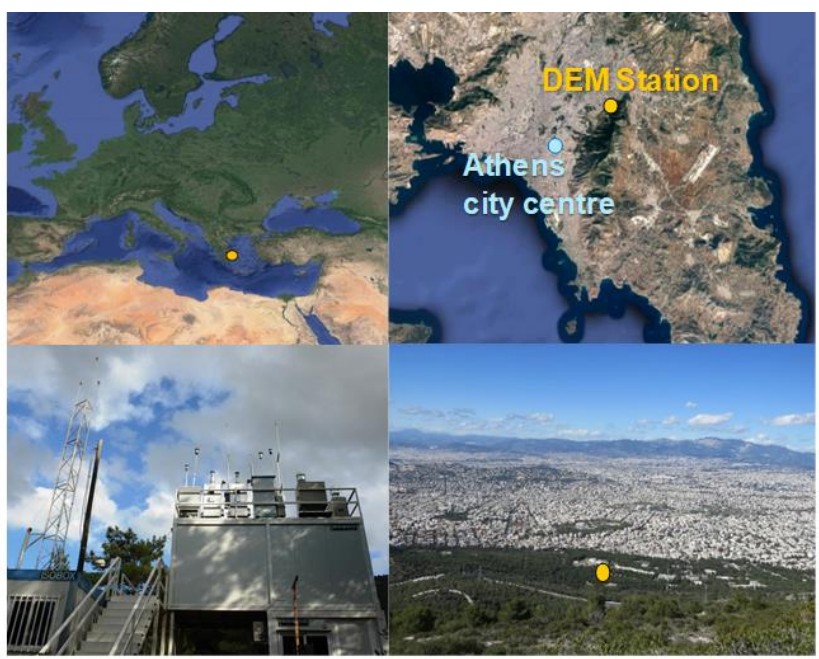


**Figure 1.** The Demokritos Atmospheric Aerosol Measurement station in Ag. Paraskevi, Athens, Greece, DEM (GAW, ACTRIS). The maps were obtained from © Google Maps (maps.google.com) Imagery 2021 Terrametrics, Mapdata 2021 and modified by the authors.

**2.2 Instrumentation**

Measurements of mass concentrations of non–refractory species (NRS) of $PM_1$ (i.e. organic; sulfate, $SO_4^{2-}$; nitrate, $NO_3^-$;

ammonium, $NH_4^+$; chloride, $Cl^-$) were performed from November 2017 to October 2018 by a Time–of–Flight Aerosol Chemical Speciation Monitor (ToF-ACSM) (Aerodyne Research Inc., Billerica, MA, USA). The ToF-ACSM was operated with a time resolution of 10 min and the data were afterwards averaged to 30 min, resulting in detection limits lower than 0.2 µg m$^{-3}$ (Ng et al. 2011). A detailed description of the instrument's main components and principle of operation is given by Fröhlich et al. (2013). In short, the instrument sampled dried (RH < 40 % with a Nafion drier) ambient aerosol through a

$PM_{2.5}$ virtual impactor. The aerosol entered the instrument through the inlet system which consists of an automatic three way valve switching system (i.e. a filter was interposed, every 20 sec, into the flow of ambient air to the instrument to measure the background signal), a critical orifice (i.e. sample flow 85 mL min$^{-1}$), and the aerodynamic lens. The aerodynamic lens focuses the submicron aerosol particles in a narrow airbeam into the vacuum chamber, at the end of which the particles impact on a heated (600 °C) tungsten plate. There, the non–refractory species are flash vaporized and subsequently ionized

by electron impact (EI) in a tungsten filament at 70 eV and detected, following their mass–to–charge ratios, by the Tofwerk time–of–flight mass analyser (ETOF).



The Relative Ionization Efficiencies (RIEs) used for organics, $NO_3^-$ and $Cl^-$ were 1.4, 1.1 and 1.3, respectively (Fröhlich et al., 2013). While the RIE values for $SO_4^{2-}$ and $NH_4^+$ were calibrated during the campaign and were found to be 1.2 and 3.4, respectively. Additionally, a collection efficiency (CE) correction factor was applied to all ACSM data, to compensate for particle losses during their collection. Generally, CE varies with chemical composition, acidity and water content of the particles (Matthew et al., 2008). However, in the present study we applied a constant CE value of 0.5 to account for the particle losses, given that the ammonium to nitrate fraction was found below 0.4 for the majority (99.9 %) of the data points (Middlebrook et al., 2012).

The equivalent black carbon (eBC) mass concentrations were also measured by an aethalometer AE33 (Magee Scientific Corp., Berkeley, CA 94703, USA). AE33 provides a real–time compensation for multiple scattering in the filter matrix and loading effects using the DualSpot Technology (Drinovec et al., 2015). Afterwards, the fraction of fossil fuel and wood burning eBC (eBCff and eBCwb, respectively) was determined as described in the Supplement (Sect. S1). In addition, the elemental carbon (EC) and organic carbon (OC) mass concentrations were measured by the thermo–optical transmittance method (OC/EC Semi-Continuous Field Analyzer, Sunset Lab, Inc.). The instrument collected aerosol samples on a 3 h basis from a $PM_{2.5}$ cut–off inlet and a flow rate of 8 l m$^{-1}$. The sampling inlet was equipped with an activated carbon denuder for the removal of organic gases from the air stream (Diapouli et al., 2017). The sample analysis was performed applying the EUSAAR2 thermal protocol (Panteliadis et al., 2015). Moreover, a high energy, polarization geometry energy–dispersive XRF spectrometer (Epsilon 5 by PANAnalytical, Almelo) was used for analysis on $PM_{2.5}$ filters, which measured the following elements: Na, Mg, Al, Si, S, Cl, K, Ca, Ti, V, Cr, Mn, Fe, Co, Ni, Cu, Zn, Br, Sr and Pb (Manousakas et al., 2017). Nitrogen oxides (NOx) and ozone ($O_3$) measurements with a 1 hour time resolution were obtained from the air quality monitoring station of the Greek Ministry of Environment and Energy air quality network located at the grounds of NCSR Demokritos campus. Standard meteorological parameters (T, Solar radiation, RH, wind speed and wind direction) were recorded at an hourly time interval. The meteorological sensors were installed on a meteorological mast, at 10 m height above ground.

## 2.3 Positive Matrix Factorization (PMF)

The data derived from the ToF-ACSM were analysed using the Aerodyne software Tofware version 3.2. The concentration in nitrate–equivalent mass and the error matrices of each species were exported from Tofware for further assessment via the PMF model. The method was implemented within the Source Finder Pro software package (SoFi Pro, Datalystica Ltd, Villigen, Switzerland) that uses the multilinear engine ME–2 (Paatero 1999) as a PMF solver (Canonaco et al., 2021). PMF is a bilinear model used to describe a non–negative matrix X using two factors (G and F), while there is also a residual matrix (E) containing the data that could not be described with G and F (Eq. (1)). For our data, the matrix X is the mass spectra of organics or total NRS through time, G is the time series of each factor and F is the matrix of the factors profile as described by Eq. (1):

$$X = GF + E, \tag{1}$$





The aim of this model is to find the minimum of the quantity Q which is the sum of the square of the ratio of the residuals (e) to the uncertainties (σ) of all the X matrix data as given by Eq. (2):

$$Q^m = \sum_{i=1}^{m} \sum_{j=1}^{n} \left(\frac{e_{ij}}{\sigma_{ij}}\right)^2, \tag{2}$$

Where m is the number of rows of F and n is the number of columns of the matrix G. The minimization of this quantity ensures that data points with low signal–to–noise ratio ($\frac{e_{ij}}{\sigma_{ij}} \ll 1$) are taken less into consideration.

Partially constrained G and/or F matrix, or a–value approach, is one of the techniques used in order to cope with the model's rotational ambiguity, which is the potential of F and G matrices to rotate, giving thus a very high number of solutions. The a–value represents the value to which the solution is supposed to vary from a reference value as shown in Eq. (3) and Eq. (4):

$$f_{j,solution} = f_j \pm a \cdot f_j, \tag{3}$$

$$g_{i,solution} = g_i \pm a \cdot g_i, \tag{4}$$

Where $f_j$ and $g_i$ are rows and columns of the matrices F and G respectively.

An important feature of the SoFi Pro software is that it enables the user to apply specific or random a–values to constrain the input profiles and/or time series with auxiliary reference data (Canonaco et al., 2013). Moreover, SoFi supports the downweighting of the data for which the signal to noise ratio is low, in order to minimize their effect on the solution. To

assess the statistical uncertainty resulting from the changes in factor profiles, a resampling strategy is usually applied in PMF modelling, called bootstrapping (Efron, 2000). This uncertainty is estimated based on variations of the obtained factor profiles coming after the rearrangement of the original input that generates a new set of initial matrices at each iteration. SoFi Pro includes the rolling window technique application; a technique that allows the user to track the variability of the factors by applying a window with selected length (usually 7, 14 or 28 days, depending on the size of the studied dataset)

that moves with a chosen step and simulations take place in that moving span providing the temporal changes in both profile and time series of the factors (Canonaco, 2021).

**2.4 Wind air mass trajectory analysis**

To investigate the potential location of NRS emission sources, wind and air mass backward trajectory analysis was performed. The conditional probability function (CPF) was used to provide directional information concerning the major

sources of NRS species. The CPF calculates the probability that in a particular wind sector and wind speed interval, the concentration of a species is greater than some specified value, which is usually expressed as a high percentile of the species of interest (e.g.75[th] percentile). In the present study, CPF analysis was performed by using the OpenAir software (Carslaw and Ropkins, 2012). A wide range of percentile values was examined to get a more complete insight into the sources of each species and each factor.





To assess the potential influence of long range transport events to NRS concentrations, the air mass backward trajectories were calculated using the NOAA Hybrid Single–Particle Lagrangian Intergrated Trajectory (HYSPLIT–4) model (Draxler and Hess, 2004; Stein et al., 2015). The 120 h backward trajectories were computed every 1 hour, at a height level of 1000 m Above Ground Level (AGL), and then further analyzed using ZeFir v3.7 (Petit et al., 2017) for the identification of the potential aerosol sources from the Potential Source Contribution Function (PSCF). The trajectories were computed using the

Global Data Assimilation System (GDAS) meteorological dataset.

## 3 Source apportionment

### 3.1 Data analysis

In the present study, two different PMF analyses were performed to apportion the sources of organic and inorganic aerosol. The first analysis included only the mass spectra of the organic aerosol (organic aerosol matrix), whereas in the second

analysis the mass spectra of organic and inorganic aerosol (combined matrix) were combined into PMF analysis, in order to investigate the sources and dynamic processes of non–refractory $PM_1$ aerosol.

For the deconvolution of the sources of the total NRS, the organic and inorganic variables and error time series matrices were exported from Tofware for each species separately (org, $SO_4^{2-}$, $NO_3^-$, $NH_4^+$, $Cl^-$) without applying RIEs or the CE correction, with a time resolution of 10 min, which was then averaged to 30 min. In order to create the combined matrix, the

variables of the inorganics that are characteristic for each species were added to the organics matrix; that is $m/z$ 18, 32, 48, 64, 80, 81 and 98 for $SO_4^{2-}$, $m/z$ 30 and 46 for $NO_3^-$, $m/z$ 16 and 17 for $NH_4^+$ and $m/z$ 35 and 36 for $Cl^-$. The variables of inorganic species selected as representative of each species, were perfectly correlated with the respective species ($R^2 \approx 1$) and accounted for the major fraction of their total mass concentration (> 76 %). After applying PMF analysis, the mass concentration of each species was calculated based on the contribution of the variables included in the initial matrix to the

total mass concentration of each species. The error values for each inorganic species were downweighted before PMF analysis by a factor of sqrt (N) (Ulbrich et al., 2009), where N is the number of ions of each species that are duplicate according to the fragmentation table (Allan et al., 2004).

In order to correct the results from nitrate–equivalent to real mass concentrations the RIEs and CE needed to be applied. This took place easily in the case of the organic matrix, by dividing the respective variables with the CE (0.5) and RIE of organics

(1.4). In the case of the combined matrix though, the factors contained more than one species, so the application of the RIEs became more complicated. The time series of each factor were decomposed to the time series of all the variables that constitute each factor using the SoFi interface. Then, at each variable, the RIE of the respective species that this variable belongs to is applied, as is also the CE and afterwards, the time series of these variables are added to form the initial factor time series.



## 3.2 PMF analysis and factor identification

The first step for source apportionment was to perform PMF analysis on winter months (November–February) in order to identify the number of factors for each matrix. For this purpose, unconstrained winter simulations took place examining a broad number of sources (3–12 factors). To identify the optimum number of factors, the slope of the Q/Qexp plot was examined, as well as the residuals of the diurnals and of the factors profiles. This resulted in a five factors solution for the organic aerosol matrix and a seven factors solution for the combined matrix. As previously mentioned, the OA matrix was described by a hydrocarbon–related factor (HOA), a cooking aerosol (COA), a biomass burning OA (BBOA) and two secondary OA (MO-OOA and LO-OOA). The combined organic–inorganic matrix was best described with the same primary factors, with two secondary inorganic factors (ammonium sulfate and ammonium nitrate) and two secondary aerosols (MOA and LOA). The next step for both analyses was to study each season separately, applying random a–values from 0 to 0.5 to the constant profiles of the primary factors and enabling the bootstrap technique for one hundred simulations in order to assess the uncertainties and check the stability of the solution obtained. Specific criteria were applied to the organic matrix to select the environmentally reasonable solutions (Chen et al., 2021), which are summarized in Table S1. Specifically, the correlation between HOA and eBCff was used as a criterion for this factor and for BBOA its correlation with eBCwb was used as well as the variation of $m/z$ 60, 73 and 115 explained by this factor. A t–test was also used for these criteria and a p–value lower than 0.05 was chosen; more details of the t–test were introduced in the Supplement of Chen et al. (2021). In the case of COA factor, the ratio of COA concentration at noon (14:00 local time) over COA mass concentrations in the morning (average COA concentration between 09:00 and 10:00 local time) was chosen to be larger than 1. Concerning the SOAs, the fractions $m/z$ 43 and 44 were monitored and should be higher than 0.

PMF analysis was then conducted on a subset of data defined by a small window of 14 days that is moved in 1 day increments across the entire dataset and as such allows capturing variations of the factor profiles (Canonaco et al., 2021). This resulted in a total of 17200 runs. The mass spectra of the three primary factors obtained through the seasonal PMF simulations were used afterwards as reference profiles for the rolling window runs. In order to select the best solutions, the same criteria for selection were used as described before (Table S1). The averaged solution for both organic aerosol and combined matrix analyses are further discussed in Sect. 4.

It needs to be mentioned that in the combined matrix analysis, for the deconvolution of organic and inorganic sources, the primary OA factors and the two inorganic factors (ammonium sulfate and ammonium nitrate) were constrained with the respective factors identified in winter under the unconstrained simulations, for both seasonal and rolling PMF simulations. Overall, tighter constraints than those applied in the OA modelling were used, allowing for a maximum variability from the anchor profiles of up to ± 20 % (random a–value 0–0.2). The criteria used in this case were the same as that of the OA interpretation that appears in Table S1.





### 3.3 Evaluation of solutions

The assessment of the uncertainty and the errors of the OA factors retrieved is a key for evaluating the solution obtained, since PMF is considered as a user–dependent technique. One way to assess the uncertainty of the retrieved solution is through the evaluation of the scaled residuals. For 99 % of the data the scaled residuals of the organic fraction PMF analysis

were in the range of ±3, which is the suggested reasonable range (Paatero and Hopke., 2003). This percentage was reduced to 91 % for the combined analysis, although it remained high enough to assume the model was well fitted. The points at which the scaled residuals exceed these thresholds (±3) were associated with peaks at OOAs and oxidized aerosols in OA and combined PMF analysis respectively, which was expected since these factors are linked to higher uncertainties due to being unconstrained.

Additionally, the PMF errors modelled for the retrieved factors, which were represented as the slope of the linear fit of the interquartile to median mass concentration values for each factor, were studied. Overall, higher error values were observed for SOA than POA factors, given their profiles perform higher variability. The combined PMF analysis was in general associated with lower PMF errors than the OA matrix analysis for all factors (e.g. 9.5 % for combined HOA instead of 10 % for OA HOA, 5 % for combined COA instead of 7 % for OA COA, 3 % for combined matrix BBOA instead of 5 % and 13

% on average for oxidized aerosols rather than 19 % for SOAs). In both cases, however, the errors were low enough to claim that we have arrived at optimal solutions.

### 4 Results and Discussion

### 4.1 Fine aerosol chemical characterization

### 4.1.1 Seasonal and diel variability

In Fig. S1 the time series of the NRS derived from the ToF-ACSM are presented. All time–related plots are in local time. The time series imply strong temporal variation of the NRS mass concentration. The maximum 30 min average total NRS concentration recorded during this campaign was 61.6 µg m$^{-3}$, while, overall, the total NRS concentration was higher than 5 µg m$^{-3}$ for 74 % of the period studied. As highlighted in Table S2, the organic fraction, as well as the total NRS concentration, presented higher values in spring and summer rather than winter at our site. This was probably connected to

the location characteristics of our site, which is located at the suburbs of Athens. In winter, higher contribution of the anthropogenic emissions (e.g. BBOA) to the total organic mass concentration was observed compared to spring and summer. In the latter case, the absence of precipitation and the increased production of biogenic volatile organic compounds in the forest near our station (Lappalainen et al., 2009) combined with enhanced photochemical activity may have led to higher formation of SOA, consequently resulting in higher organics concentrations (Tables S3, S5). Simultaneously, sulfate levels

were higher in spring and summer. Apart from the enhanced photochemistry, the increase in height of the boundary layer and the atmospheric dynamic conditions, favoured the regional and long–range transport and mixing of polluted air masses,



resulting in enhanced sulfate concentration (Cusack et al., 2012; Dayan et al., 2017). This is in agreement with other studies performed in Athens (Theodosi et al., 2018; Stavroulas et al., 2019). HYSPLIT was used to track the different seasonal long–range origins of sulfate. As appears in Fig. S2, in spring, summer and early autumn sulfate originated mainly from central

and eastern Europe and also Turkey, which are known to carry a significant amount of industrial emissions (Karagulian et al., 2015). This effect was observed as a maximum value of sulfate concentration for these seasons. Nitrate, which is semivolatile, presented, as expected, higher concentrations in the cold months when the lower temperature favours the formation of ammonium nitrate and portioning nitrate into the particle phase, instead of the gas phase in which it appears when temperature rises and nitrate is predominantly in the $HNO_3$ form (Lin and Cheng., 2007). Ammonium presented higher

concentrations in warmer months, following a similar pattern as that of sulfate. Chloride exhibited its highest concentration in winter correlating its emission to biomass burning, as was also resolved by the unified matrix PMF analysis discussed in Sect. 4.2.2.

In comparison to the results reported in a previous study conducted in the centre of Athens (Stavroulas et al., 2019) by the National Observatory of Athens (NOA), the suburban site presented lower concentrations for all NRS in wintertime, which

is mainly attributed to the higher anthropogenic emissions generally occur in the urban area of Athens. Specifically, we observed that all NRS presented concentration levels 3 to 4 times lower than those at the centre of Athens in winter except for sulfate for which a lower difference in the concentration levels was observed. For the rest seasons, the inorganic species presented similar concentration levels at both sites (i.e. higher contribution of secondary aerosol than primary emissions to the observed NRS levels), while organics were higher at the suburban site in spring and autumn. This is probably related to

the enhanced SOA formation at the suburban area which is dominated by pine tree vegetation. Black carbon exhibits lower concentrations in the suburbs, in agreement with the comparison study conducted by Kalogridis et al (2018).

Figure 2 presents the daily variability of NRS species for each period studied. Nitrate always increased during the night, however, in winter and spring a distinct morning peak also appeared which is probably linked to photochemical activity, meteorological conditions (gas–to–particle equilibrium) and ammonia availability. On the other hand, the diurnal profile of

sulfate was closely related to its regional nature, with the main peak occurring either during the night (summer) or in the afternoon and early evening (winter and spring) reflecting the changes to the boundary layer height. Sulfate's seasonal diurnal variability agreed quite well with that of ammonium. Chloride presented two distinct peaks (i.e. morning and late afternoon) for all the seasons, related to temperature–dependent gas–particle partitioning of chlorine (i.e. chloride is primarily detected as ammonium chloride), biomass burning emissions and prevailing atmospheric conditions. The organic

fraction showed a midday/early afternoon and an evening peak. Its diurnal cycle was most likely a combination of primary emissions from various sources, and secondary aerosol formation during the day. It has to be noted that all NRS species appeared to have increased concentrations during the night which may also be attributed to the increased atmospheric stability during the night (shallow nocturnal boundary layer).

On average, during the period of this campaign particulate matter consisted of 51.3 % organics, 34.7 % sulfate, 9.4 %

ammonium, 4.4 % nitrate and 0.2 % chloride. In Fig. S3 the wind rose plots for each season appear, while Fig. S4 presents





the seasonal bivariate CPF polar plots for all NRS to investigate the potential source regions of these species. The polar plots for organics showed highest concentrations for low and moderate wind speeds near the center area as well as in SE and NE directions indicative of both local emissions and regional transport. High concentrations of sulfate were observed for low, moderate and high wind speeds from the SE sector, suggesting that a combination of local emissions and regional and long–

range transport may significantly contribute to the observed sulfate levels. This was also the case for ammonium, underlying the common origin of these species. Nitrate was primarily locally produced. The high potential source region of particulate nitrate coincided with the one observed for NOx, linking the particulate nitrate with the traffic–related NOx emissions (vehicle exhausts). High concentrations of chloride were observed at relative low wind speed implying that it was rather locally emitted, probably linked to local secondary aerosol formation and biomass burning emissions, as will also be

discussed in Sect. 4.3.

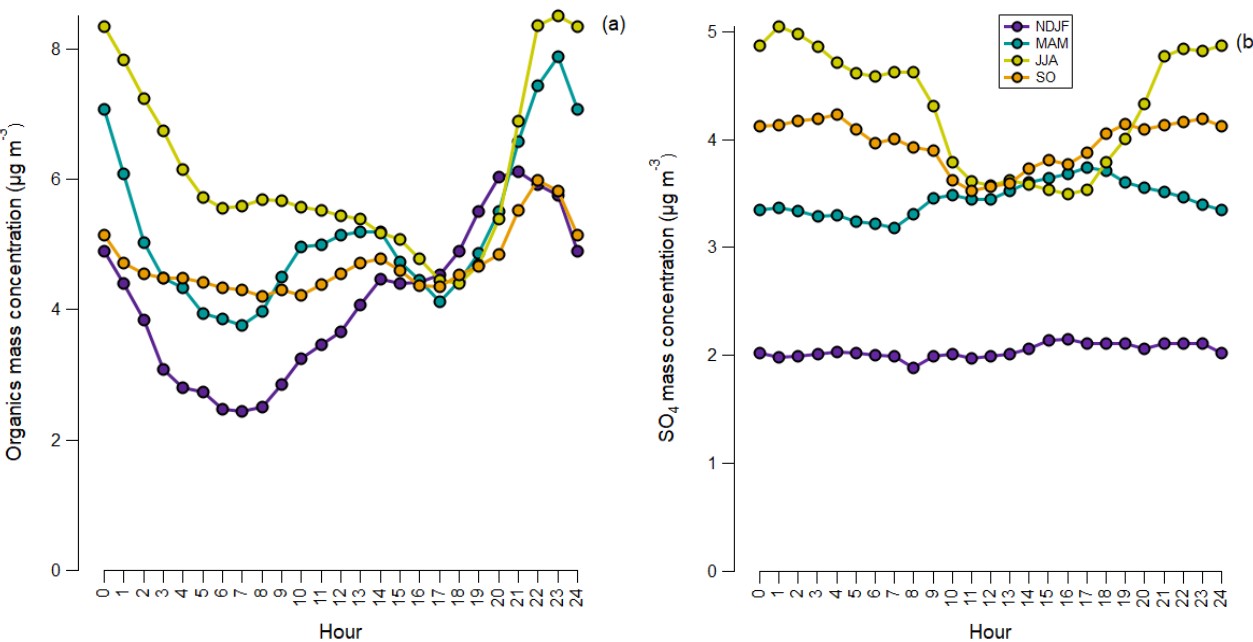





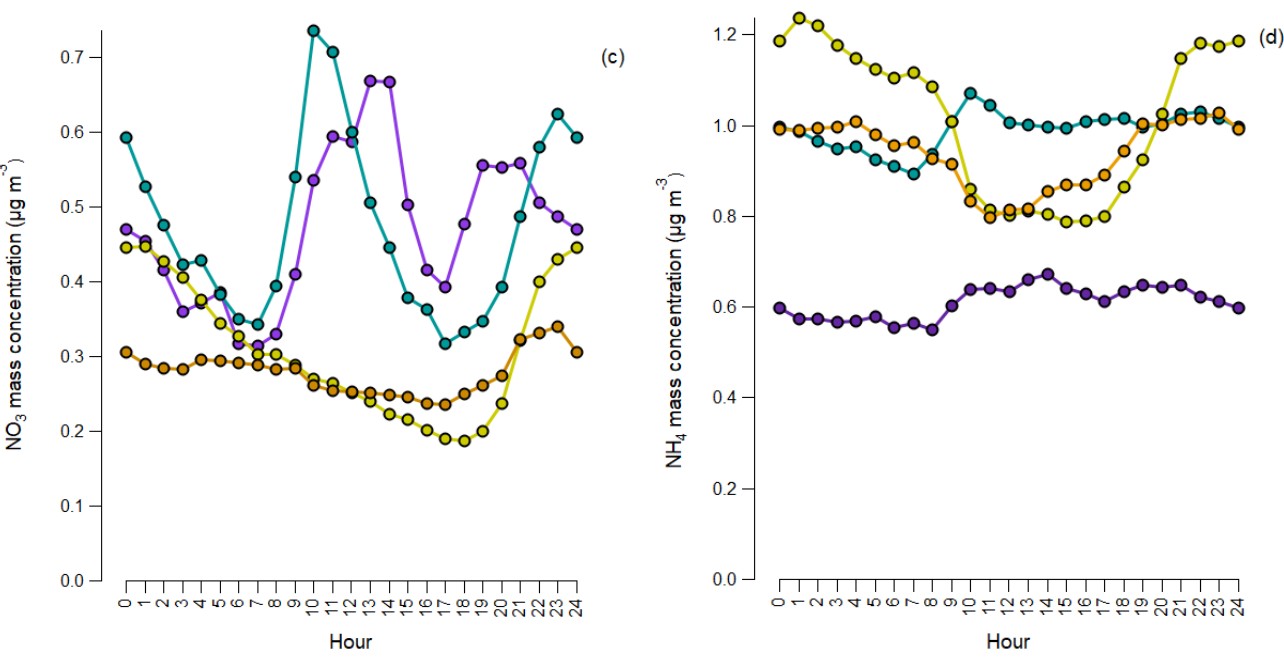


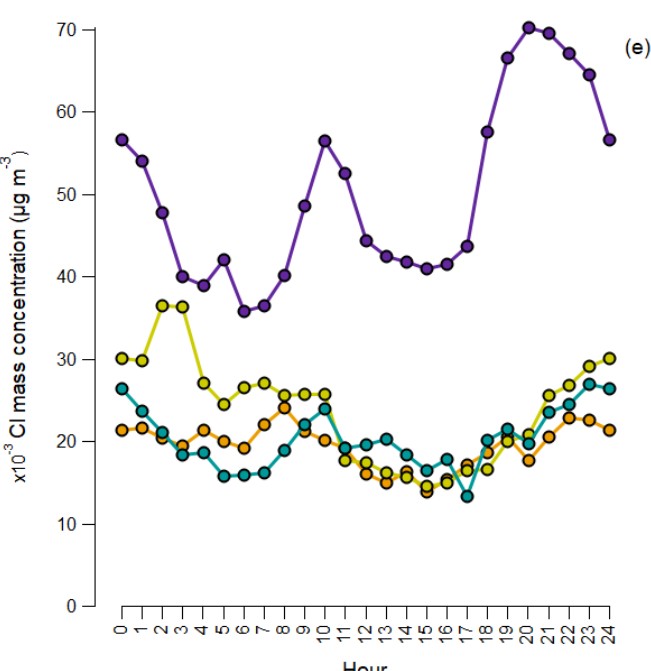

**Figure 2.** Mean diurnal variation of hourly concentrations of organics (a), sulfate (b), nitrate (c), ammonium (d) and chloride (e) for each period: November–February (NDJF), March–May (MAM), June–August (JJA), September–October (SO) in local time.



### 4.1.2 Data from collocated instruments

Figure S5 depicts the time series of supplementary data from collocated instruments; that is eBC, NOx, O$_3$, EC/OC, temperature, relative humidity, wind speed and wind direction. As shown, equivalent BCff presented a quite stable contribution to ambient particulate matter throughout the year, while eBCwb had a stronger influence in winter as expected due to biomass burning emissions related to residential heating. During the study period, the cold months (October–March) were characterized by high humidity (74 % mean) and relatively low temperatures (12.7 °C mean), while the warm months

(May–September) were characterized by moderate humidity (54 % mean) and temperature (24 °C mean) levels. The prevailing wind speed and direction are presented for each season. As highlighted also in Fig. S3, winter was influenced mainly by southwestern winds, implying a significant contribution from the emissions related to anthropogenic activities from the city centre. In spring the air masses originated mainly from southwest (urban city centre) and southeast (downslope wind) directions, possibly associated with Saharan dust events as well. In summer, the wind originated mainly from the

southeast and northeast directions, underlying the role of regional transport along with the local aerosol emissions and formation processes, while in September and October the measurement site was mainly affected by north-western wind directions (regional and long–range transported aerosol).

### 4.2 Source apportionment of organic aerosols

The mass spectral profiles of the five factors up to mass to charge ratios of 100 are presented in Fig. 3a, along with the time series (Fig. 3b) and the diurnal variation patterns (Fig. 3c) of the respective mass concentrations. The remaining part of the mass spectrum is depicted in Fig. S6. In Fig. S7a, the polar plots present the spatial distribution for each factor, while in Fig. 7b the polar plots for external data (eBCff, eBCbb, NOx and O$_3$) are depicted. The relative contribution and actual mass loadings for each factor and season are summarized in Table S3. The mass spectra, diurnal variations and potential sources

of these factors will be discussed in detail in the following sections.



**Figure 3.** Mass spectra (a), time series (b) and diurnal trends (c) of the five organic aerosol factors.



*Hydrocarbon–like organic aerosol*

The hydrocarbon–related (HOA) factor, linked to fossil fuel combustion emissions, was distinctly recognized and attributed to the traffic–related emissions from the urban area of Athens, that are transported to the measurement site under westerly wind directions (Fig. S7a). From the plots in Fig. S7b it can be seen that HOA shared the same emission origin with NOx
and eBCff. The HOA profile was dominated by peaks characteristic of the alkanes $C_nH_{2n-1}^+$ and $C_nH_{2n+1}^+$, with high contribution of *m/z* at 27, 41 and 55 ($C_nH_{2n-1}^+$) and 29, 43 and 57 ($C_nH_{2n+1}^+$) (Zhang et al., 2005). The fingerprint of the traffic–related factor profile has been identified to be almost stable over spatially different sites across Europe (Crippa et al., 2014), although strongly dependent on the vehicle and fuel type commonly used. For example, diesel fuel and lubricant oil aerosols are dominated by ions series typical of normal and branched alkanes, but also by series typical of cycloalkanes and
aromatics (Canagaratna et al, 2004). In the present study, the unconstrained HOA profile obtained was highly correlated HOA profile from Crippa et al (2013) ($R^2$ = 0.98), which is typically used to constrain the HOA profile in urban environments.

HOA factor presented an overall good correlation with the time series of fossil–fuel combustion indicators like NOx (R–Pearson = 0.69), eBCff (R–Pearson = 0.69) and EC (R–Pearson = 0.58) as shown in Table S4. The good agreement between
the HOA and the traffic–related external parameters validated the correct identification of HOA profile and the accurate separation of COA from HOA. In general, the peaks observed at the time series of this factor coincided with peaks also observed in the respective external data time series, either eBCff or NOx, except for the peaks observed on the 8[th] of February (connected to cooking emissions), and at the end of March, which both will be discussed in detail below. The diurnal variability of HOA's mass concentration presented two peaks, one morning peak at 9:00 and one evening peak at
21:00 (local times), coinciding with the morning rush hours and the evening traffic emissions and the shallow nocturnal boundary layer. As summarized in Table S3, HOA's average contribution to the total OA was 15 %, with its seasonal contribution at 18 % for winter and spring decreasing to 10–13 % in summer and early autumn. The latter decrease in HOA mass concentration and OA contribution was expected, since traffic–related emissions are reduced during the summer in Athens (Stavroulas et al., 2019), while SOA formation is enhanced.

*Cooking emissions*

The mass spectrum representative of cooking emissions was also identified in our study. The chemical fingerprint of the COA factor profile was similar to HOA's, in terms that the same variables that dominated the HOA profile were also present in the COA emissions profile, but the *m/z* at 55 was the prevailing one. From previous studies the difference in chemical identity between the variables 55 and 57 assigned to HOA and COA factors has been identified (Mohr et al., 2009). The
variable at *m/z* 55 in traffic emissions mass spectra corresponds to 2-methylprop-1-ene ($C_4H_7^+$), while the respective variable coming from cooking emissions is related to 2-methylprop-1-ene and prop-2-enal ($C_3H_3O^+$). The variable at *m/z* 57 associated with traffic emissions is a tracer of butane ($C_4H_9^+$), while in the case of cooking emissions *m/z* 57 is a tracer of





butane and prop-1-en-1-ol ($C_3H_5O^+$), representing oxidized fatty acids that are main tracers of cooking emissions (Crippa et al., 2014).

In a study conducted in our station deploying an AMS for a short–term campaign in summer 2012, no cooking emissions were identified, instead two HOA factors were retrieved, one of which was interpreted as mixed traffic and cooking source (Kostenidou et al., 2015). In the present study, the identified COA factor had surprisingly similar characteristics both in terms of the mass spectrum profile as well as the diurnal pattern of COA mass concentration, with the fresh COA factor retrieved in a previous study conducted in Greece (Kaltsonoudis et al, 2017). On the 8th of February, a distinct peak in COA

was observed that was related to the barbeque festival "Smokey Thursday". The simultaneous peak observed at that day in HOA time series was attributed to the enhanced organic aerosol emissions during this event that impedes the model from configuring right the factors. The diurnal variability pattern of COA mass concentration presented a bimodal pattern (Fig. 3c), with the two peaks coinciding with lunch and dinner time. The OA loading from cooking emissions over the day was lower than the respective from traffic, with a total duration of 8 h. COA's seasonal contribution to total OA followed the

same trend as HOA, decreasing from 19–21 % in colder months to 14–16 % in warmer months, while the average contribution of this factor was 17.7 %. As shown in Fig. S7a, this factor had a local character, linked to the cooking emissions originating from the urban environment in close proximity to the measurement site.

*Biomass burning organic aerosol*

We were able to resolve a factor dominated by wood burning (*m/z* 60 and 73) and PAH (*m/z* 77, 91, 115, 128, 165, 167)

tracers. In order to ensure that the PAHs presented in the BBOA profile were attributed to biomass burning (Li et al., 2009), we conducted PMF runs constraining the profile of our BBOA factor with the BBOA profile retrieved from another study (Ng et al., 2011), trying to resolve a PAH–related factor in case it exists; no environmentally reasonable solution could be reached. Additionally, since PAHs can be also generated by gasoline car exhaust emissions (Okuda et al., 2010), we constrained our HOA and BBOA profiles up to the variables at m/z 100 and conducted 100 simulations. Afterwards, using

the criteria list, we eliminated the runs in which the PAH–related variables were attributed to HOA instead of BBOA and found out that for more than 70 % of the simulations these variables were associated to BBOA. Other PAH sources may include coal combustion (Okuda et al., 2010), but coal is generally not used in Greece for heating purposes, while the correlation of this factor with industry–related markers measured by XRF analysis on $PM_{2.5}$ filters was very low (R–Pearson < 0.2 between BBOA and Pb, Cu, Mn, Zn, Sn, Cr, Cd, Rb, S, Fe, V, Ni). Moreover, polar plots revealed no connection

between this factor and port emissions (Fig. S7). Thus, we concluded that this factor indeed originated primarily from biomass burning.

The time series of this factor were very relevant to the time series of the wood burning fraction of eBC (eBCwb) obtained from the aethalometer (R–Pearson = 0.74), as illustrated in Table S4. The strong dependence of the concentration of biomass burning to the temperature is also depicted in Fig. 3b, where it is clear that the increased contribution of BBOA to OA

concentrations generally occurred at low temperature (wintertime). However, the peaks in the time series of this factor were





also connected to wild forest fires, like the one that occurred on the 23$^{rd}$ of July at the region of Attiki. In any case, all these peaks observed in BBOA time series were also confirmed by peaks in the eBCwb time series. The winter contribution of BBOA to OA mass concentration was close to 18 % in winter and decreased to 5 % in summer (Table S3). BBOA's spatial distribution (Fig. S7a) confirmed the strong local character of this factor, although long–range transport from the North sector may also have contributed to the increased BBOA levels.

*Oxygenated organic aerosols*

The oxygenated organic factors retrieved in the current study were of two types: one more oxidized oxygenated organic aerosol (MO-OOA) and one less oxidized (LO-OOA). Oxygenated organic aerosols (OOA) have as main tracers the *m/z* variables at 28, 29, 43 and 44. MO-OOA profile was dominated by *m/z* 44 (corresponding to the $CO_2^+$ ion) instead of *m/z* 43; the fraction at *m/z* 44 (*f*44) provides information regarding the degree of oxygenation of the respective factor. On the other hand, LO-OOA mass spectrum was represented by almost equal contributions of *m/z* 43 and *m/z* 44 ($C_2H_3O^+$) (Ulbrich et al., 2009). LO-OOA was significantly affected by temperature and presented a pronounced seasonal variation pattern (Fig. 3b). LO-OOA's contribution on OA mass concentration in summer (31 %) was double that of winter (14 %) (Table S3). MO-OOA's contribution to the total OA was on average 34 % with no significant seasonal variability. LO-OOA exhibited similar correlation with the three inorganic ions ($SO_4^{2-}$, $NO_3^-$ and $NH_4^+$), except for summer when it was highly correlated with $NO_3^-$. MO-OOA showed good correlation with $SO_4^{2-}$ and $NH_4^+$ in all seasons and with $NO_3^-$ only in September–October. These relationships imply the complicated internal mixing of organic and inorganic species that will be further discussed in Sect. 4.3. From the polar plots presented in Fig. S7a and Fig. S7b it can be seen that the areas where the probability of SOAs being higher were similar to that of the oxidants NOx and $O_3$. More specifically, MO-OOA originated from areas rich in both NOx and $O_3$ concentrations, while LO-OOA was mainly found on the NE and SE of our station, where $O_3$ primarily appeared. This highlights possibly different oxidation mechanisms that take place to form the two types of SOAs in our site.

*OOAs aging state*

The subtracted *f*44–*f*43 triangle plot was used to evaluate the goodness of the fit of the retrieved PMF solution (Ng et al., 2011), emphasizing on the secondary OA factors which are more inclined to change in time, since their profiles were not constrained given that different SOA formation mechanisms may result in quite different mass spectrum profiles. As described in Sect. 3.2, the rolling window approach that was implemented here allows for the temporal evolution of the SOA factors to be further investigated through the subtracted *f*44–*f*43 plots. In Fig. S8 the fraction *m/z* 44 to the fraction *m/z* 43 plots, after the subtraction of the contribution of the primary factors (Canonaco et al., 2015), are shown for each season (markers are colour coded by concentration). The rectangular and circular markers (colour coded by date) represent the LO-OOA and MO-OOA concentrations, respectively. It can be seen that for all seasons the two SOA factors proposed could fully describe our data as most points fell inside the triangle area (Ng et al., 2011). However, a quite different seasonal pattern was revealed for both OOA factors, according to which, winter presented a vertical arrangement between the two





SOA factors, while the shape became more horizontal and moved to the right side of the triangle in summer. During early spring (March) and October the data followed the trend of winter, while in April, May and September they followed a

pattern closer to the one presented in warm months. This behaviour was indicative of the different prevailing SOA formation mechanisms for each season.

### 4.3 Source apportionment of submicron aerosols

Figure 4 depicts the average profiles (Fig. 4a) of the seven factors resolved from the combined input matrix PMF analysis for $m/z$ up to 100. The profiles are the result of the average of 100 simulations, after applying the bootstrap technique and a

rolling window of 14 days for the PMF runs. In the same figure, the time series (Fig. 4b) and the daily trends (Fig. 4c) of each factor's mass concentration are presented, while Table S5 shows the actual mass loadings of each factor and their relative contribution to the total NRS mass concentration for each season. Figure S9 presents the mass spectrum of NRS factors for $m/z$ 100–200 and Fig. S10 depicts the CPF polar plots of the seven sources identified.

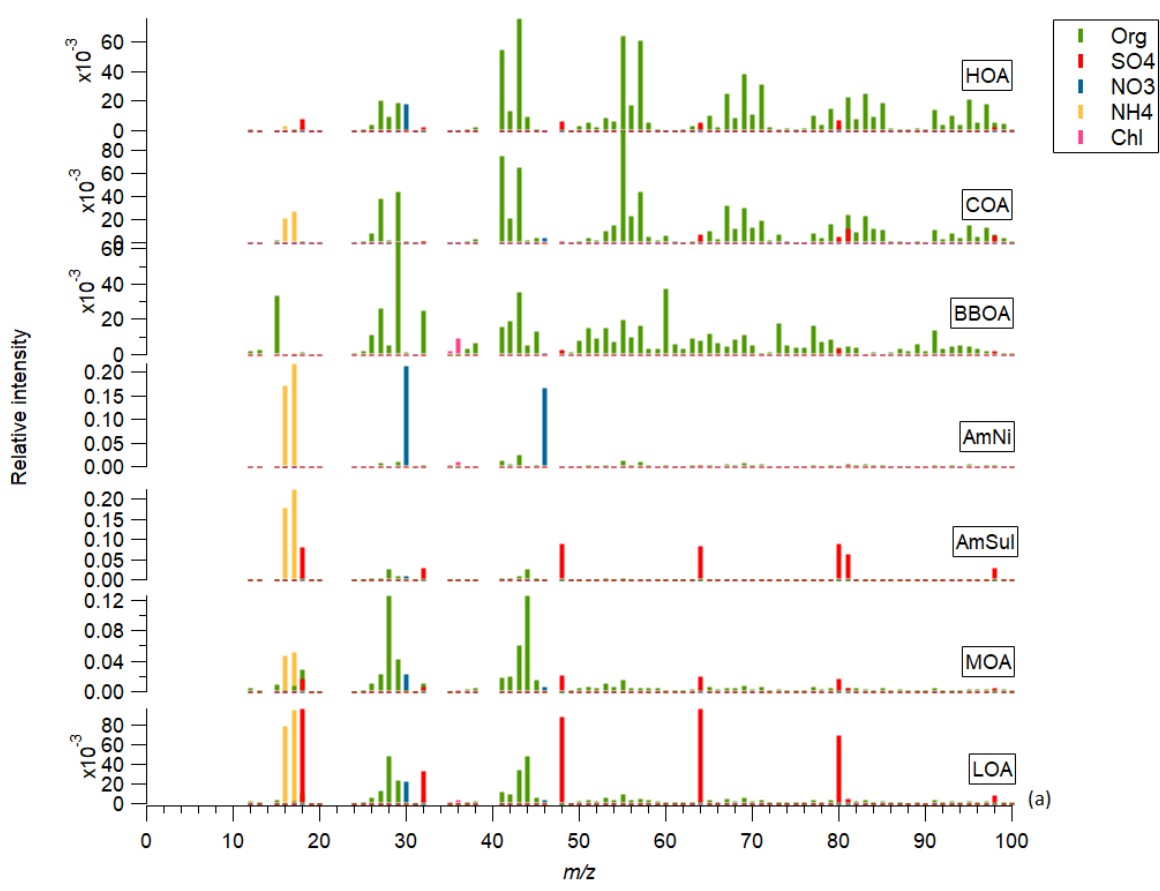


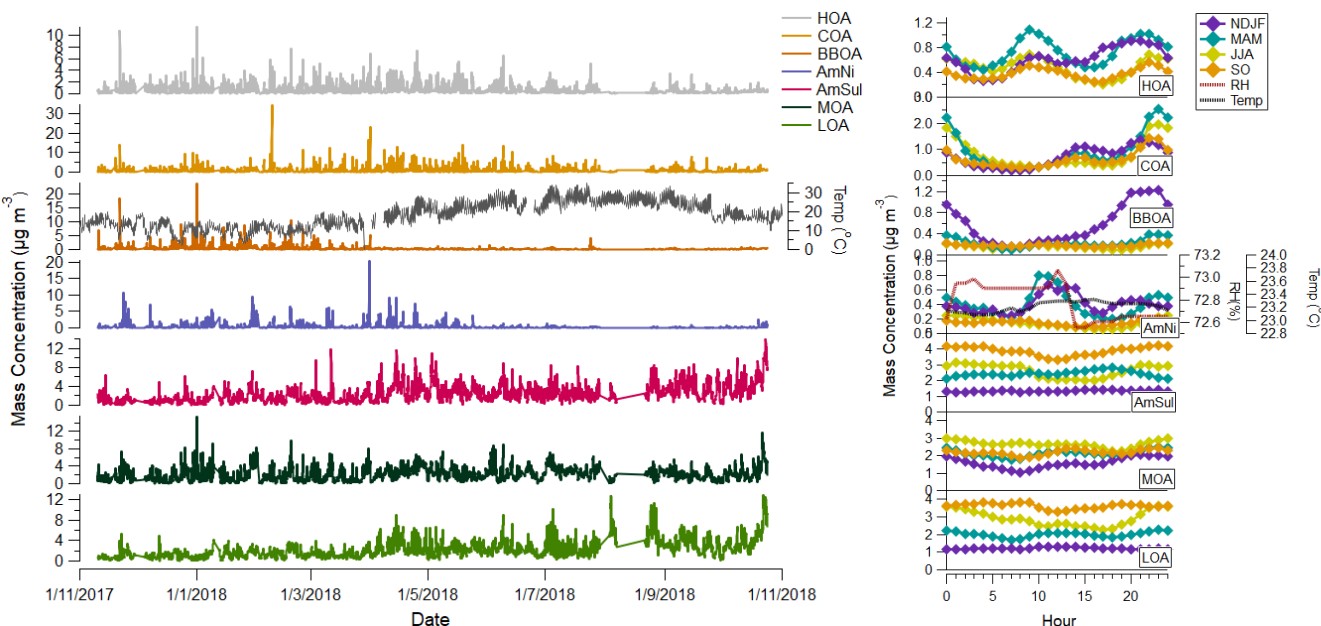

**Figure 4.** Mass spectra (a), time series (b) and diurnal plots (c) of the seven submicron aerosol sources.

*Primary Organic factors (HOA, COA, BBOA)*

The hydrocarbon–related factor that was retrieved with the combined matrix PMF method indicated the contribution of the

same *m/z* variables to the mass spectral profile of organics as the previously described HOA factor. The deconvolution of our combined organic and inorganic dataset matrix revealed a small contribution of inorganic species in this traffic–related factor. More specifically, combined HOA contained 94 % organics, 3.6 % $SO_4^{2-}$ and 2.3 % $NO_3^-$. The time series of the HOA factor obtained from the combined matrix presented good correlation with the organic matrix resolved HOA factor (R– Pearson = 0.87). The peaks of the previously resolved HOA (Sect. 4.2) that were attributed to poor separation of the OA

factors were not present in the combined matrix analysis, highlighting the improvement of the solution. The polar plot of combined HOA resembles significantly to the one from OA PMF analysis, as well as those of fossil–fuel markers (eBCff and NOx), implying the good configuration of this factor with both analyses (Fig. S10). In addition, it was observed that the correlation between this factor and external tracers of traffic emissions (eBCff, NOx and EC) improved as can be seen in Table S4 and Table S6.

Cooking–related emissions were again resolved as COA. This factor consisted mainly of organics (93.7 %) and presented low contribution of inorganic ions; $NH_4^+$ (2 %), $SO_4^{2-}$ (3.8 %) and $NO_3^-$ (0.5 %). COA's mass spectral profile resolved from combined PMF analysis resembled the previously identified one. The diurnal pattern of this factor presented again two peaks that coincided with lunch and dinner time, and its time series highly agreed with the previously resolved COA's time series (R–Pearson = 0.92). Moreover, the polar plots of COA revealed again local emissions (Fig. S10).





The factor connected to biomass burning was also identified through the combined PMF analysis. In this factor the inorganics presented lower contribution that on the other two POA factors. Combined BBOA was composed almost entirely of organics (97.8 % organics, 1 % $SO_4^{2-}$ and 1 % $Cl^-$). The two BBOAs resolved were highly correlated (R–Pearson = 0.88), while they also presented similar diurnal patterns. Again, this factor's directionality (Fig. S10) showed that it was a source affected by the city on the west and northwest of our site.

*Ammonium Nitrate (AmNi)*


The ammonium nitrate factor resolved in this study was composed of 55 % $NO_3^-$ and 18 % $NH_4^+$. The $NH_4:NO_3$ ratio was 0.33, which is close to the theoretical ratio of 0.29 for pure ammonium nitrate. The respective ratio obtained by Sun et al (2012) was 0.36, while Äijälä et al (2019) reported a ratio of 0.46. In our study, nitrate was primarily present as ammonium nitrate; this factor accounted for 81.5 % of total nitrate. Ammonium nitrate's temporal variation agreed well with nitrate's
(R–Pearson = 0.90). The diurnal variation of this factor showed enhanced concentration at noon (Fig. 4c). Nevertheless, further insight into its diurnal trend in different seasons revealed that in summer an increase in temperature caused a decrease in the concentration of ammonium nitrate, while in winter ammonium nitrate's daily concentration followed the trend of RH. This was in agreement with the behaviour of $NH_4NO_3$ in respect to the equilibrium constant value that controls the gas–particle partitioning of nitrate and depends on the temperature and the relative humidity of the atmosphere. More
specifically, if the RH is higher than the deliquescence RH (DRH) of particulate $NH_4NO_3$, then the equilibrium constant for ammonium nitrate primarily depends on RH; the increase in RH favours particulate $NH_4NO_3$ formation. Whereas, for RH levels lower than this critical point, the formation rate of $NH_4NO_3$ is inversely proportional to the air temperature (Lin and Cheng, 2007). In our case, the winter RH was higher than the DRH of $NH_4NO_3$ (62 %), thus the equilibrium constant value increased with increasing RH, as depicted in Fig. 4c. Fig. 4b highlights the effect of temperature on the equilibrium between
particulate ammonium nitrate and gas phase $HNO_3$ in summer; the increase in temperature was translated as a decrease in particulate ammonium nitrate's concentration, and consequently an increase in the formation rate of gaseous $HNO_3$. Moreover, it was observed that ammonium nitrate's peak in cold months occurred three hours after the morning HOA peak, which further indicated the formation of ammonium nitrate through the reaction of traffic–related NOx and ammonia (Fig. 4c). In warm months, on the other hand, no morning peak existed, which combined with the lower particulate nitrate
concentration levels during these months, leaded to the conclusion that the background NOx mainly participated in ammonium nitrate's formation in summertime. Generally, the pronounced peaks identified in AmNi time series coincided with peaks observed in $NH_4$ and $NO_3$ time series, as measured with the ToF-ACSM. Figure S10 further confirms that ammonium nitrate was locally formed.

*Ammonium Sulfate (AmSul)*

A factor dominantly composed of sulfate and ammonium was retrieved in this study. 64 % of the mass of this factor was attributed to $SO_4^{2-}$ and 19.5 % $NH_4^+$. The theoretical ammonium to sulfate aerosol ratio typically ranges between 0.18





(NH$_4$HSO$_4$) and 0.36 ((NH$_4$)$_2$SO$_4$). In our case, the respective NH$_4$:SO$_4$ ratio was 0.31, indicating that the sulfate presented in this factor was almost neutralized as (NH$_4$)$_2$SO$_4$ and further supporting the successful deconvolution of this factor. This factor contained 53 % of the total sulfate and it highly correlated with ACSM SO$_4^{2-}$ (R–Pearson = 0.91) (Table S6). Sun et al (2012) also retrieved a SO4–OA factor, but in that study, 18 % of the mass of this factor was organic with a high degree of oxidation (O/C = 0.69), the highest among the other factors retrieved, while the NH$_4$:SO$_4$ ratio was 0.34 which is close to the theoretical one for pure ammonium sulfate. Äijälä et al (2019) retrieved a factor of ammonium sulfate with NH$_4$:SO$_4$ ratio between 0.2 and 0.24.

*Secondary aerosols (MOA, LOA)*

Two factors representative of secondary aerosols were identified, i.e. a less oxidized, LOA (less oxidized aerosol) and a more oxidized, MOA (more oxidized aerosol), named based on the oxidation state of the organic part of these factors ($f_{44}$). Inorganic components significantly contributed to these factors. LO-OOA and MO-OOA, retrieved by PMF analysis on the organic fraction, were well correlated with the inorganic species (Table S4), implying that an intrinsic relationship between aged organic and inorganic species may exist. By applying the combined matrix PMF analysis, the mixing characteristics between organics and inorganics can be tracked. Specifically, on a yearly average, MOA consisted of 81 % organics, 11 % SO$_4^{2-}$, 4 % NH$_4^+$, and 4 % NO$_3^-$, while LOA included 41 % organics and mixed a high amount of SO$_4^{2-}$ (47.7 %), and also 3.3 % NO$_3^-$, 7.4 % NH$_4^+$ and 0.6 % Cl$^-$. Generally, the mixing of carbonaceous and sulfate compounds in the atmosphere has been established (Murphy et al., 1998; Adachi and Buseck, 2008), and since atmospheric sulfate is in most cases neutralized by particulate ammonium, the organic–ammonium sulfate mixtures are a significant part of atmospheric aerosols (Bertram et al., 2011). This was highly observed in this study as the deconvolution of the secondary aerosols yielded SOAs mixed with a considerable amount of ammonium and sulfate particles, especially in LOA.

MOA presented a strong correlation with MO-OOA (R–Pearson = 0.86) whereas LOA showed a lower but still significant correlation with LO-OOA (R–Pearson = 0.68). LOA, as well as LO-OOA, presented a strong correlation with temperature (Fig. 4). LOA's seasonal contribution to total NRS was minimum in winter (19 %), while peaked in summer (29 %) and September–October (32 %), whereas MOA's contribution to total NRS also peaked in summer to 27 % but dropped in SO to 20 %. Overall, the contribution of organics to MOA and LOA followed the same trend as that of MO-OOA and LO-OOA. While, the difference in absolute values between MO-OOA/MOA and LO-OOA/LOA concentrations was mainly driven by the sulfate apportioned to the combined secondary factors. The bivariate polar plots of MOA and LOA resembled the respective plots of MO-OOA and LO-OOA, respectively. Specifically, we observed higher concentrations of MOA and LOA for low and moderate wind speeds around the centre area as well as from NE and SE, suggesting that a combination of local and regional sources may have contributed to the observed concentrations. These regions were also associated with increased concentrations of inorganics (NO$_3$, NH$_4$, and SO$_4$) and O$_3$.

The time–dependent profiles of the rolling solution further allowed the quantification of the contribution of each species in the secondary aerosols for different seasons (Fig. 5 and Fig. S11). LOA presented very different composition in each season;



in spring and summer organics and inorganics equally contributed to the LOA's mass concentration, while in winter and early autumn inorganics presented higher contribution than organics. The apportionment of inorganic species in less oxidized OA provided an indication that less oxidized OA might be internally mixed with inorganic species (Zhang et al., 2005). The more oxidized aerosol presented a more stable composition throughout the year, with organics being the prevalent species accounting for up to 86 % of MOA concentration. Although MO-OOA showed greater mass concentration values than LO-

OOA, the secondary aerosols from combined PMF analysis presented inversed characteristics; LOA demonstrated higher values than MOA. This was attributed to the inorganic part mixed with the SOAs which was more important in the less oxidized OAs, as the organic part was higher in absolute mass in MOA than LOA (Fig. 5).

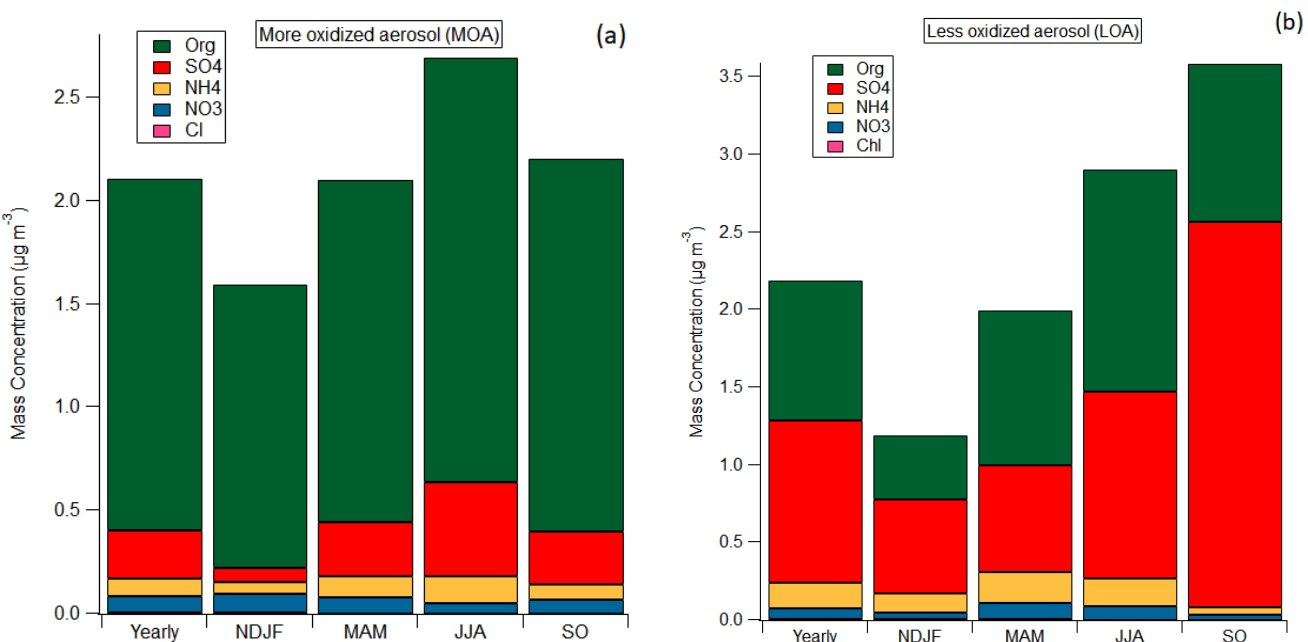

**Figure 5.** Mass concentration of each species in MOA (a) and LOA (b) in different seasons: Yearly, November–February (NDJF), March–May (MAM), June–August (JJA) and September–October (SO).

Figures S12 and S13 survey the contribution of each species in the NRS factors and the contribution of each NRS factor in the non–refractory species, respectively, in both relative (Fig. S12a and Fig. S13a) and absolute (Fig. S12b and Fig. S13b) terms. It appears that HOA, COA and BBOA consisted mainly of organics. Ammonium and sulfate together accounted for

more than 80 % of the total mass of AmSul factor, while ammonium and nitrate accounted for more than 70 % of AmNi factor with 25 % of its mass being mixed with OA. Both organics and ammonium sulfate significantly contributed to the total LOA mass concentration; organic and sulfate accounted for about 41 and 48 % of the total mass respectively, but as will be discussed below its seasonal contribution significantly varied. MOA was primarily composed of OA with 15 % of its mass consisting of ammonium sulfate. Overall, organics were present in all the factors, but they contributed less to AmNi


and AmSul. Sulfate was mainly present in AmSul factor and the secondary LOA factor. Ammonium was equally and mainly
        distributed in the two inorganic factors (AmNi and AmSul) while nitrate was primarily present in the AmNi factor. Chloride,
        although present in very low concentrations, it was equally attributed to ammonium nitrate and BBOA, highlighting the two
        main sources of chloride: secondary formation of particulate ammonium chloride and biomass burning emissions.

## 5 Conclusions

The scope of this study was the characterization and source apportionment of the organic fraction and total NRS of the
        Mediterranean city of Athens and specifically at a suburban site. PMF was successfully employed for the two different initial
        matrices; one containing the organics and one combining the organic and the inorganic species ($SO_4^{2-}$, $NO_3^-$, $NH_4^+$ and $Cl^-$)
        of a yearlong ToF-ACSM dataset. From the first analysis, five organic aerosol sources were retrieved, while combined PMF
        analysis yielded seven factors. With both analyses three primary organic aerosol factors were resolved; one hydrocarbon–
related (HOA) from traffic emissions, one from cooking emissions (COA) and one related to biomass burning (BBOA). The
        organic aerosol interpretation produced two more factors; one more oxidized (MO-OOA) and one less oxidized OA (LO-
        OOA), while these factors were mixed with inorganic species as resolved from the combined PMF analysis, named MOA
        and LOA respectively). Two additional factors identified with the latter analysis were ammonium nitrate (AmNi) and
        ammonium sulfate (AmSul). The absolute contribution of each primary factor obtained from the OA PMF analysis was 15 %
HOA, 18 % COA and 9 % BBOA, while the SOAs contribute 34 % (MO-OOA) and 24 % (LO-OOA). The factors retrieved
        from the combined PMF analysis contribute to the total NRS as follows: HOA 7 %, COA 9 %, BBOA 3 %, AmNi 3 %,
        AmSul 28 %, MOA 24 % and LOA 26 %. The primary OA factors retrieved from the combined PMF analysis were mainly
        organic (95 % on average). MOA was also mainly composed of organics with 11 %, 4% and 4% of its mass attributed to
        $SO_4^{2-}$, $NO_3^-$ and $NH_4^+$, respectively, while LOA presented a seasonal variation in its composition with the average
contribution of $SO_4^{2-}$ (47.4%) being slightly higher than that of organics (41%). External data from collocated instruments
        were used in order to identify the environmentally reasonable simulations. HOA presented in both cases significant
        correlation with traffic–related black carbon (eBCff), while BBOA correlated well with wood burning black carbon
        (eBCwb). The time series of MO-OOA and LO-OOA showed a relationship with inorganic species, which was further
        elucidated by the combined PMF analysis. MOA and LOA both showed significant presence of ammonium and sulfate in the
mass spectra, with both factors representing a prevailing class of ambient aerosols; that of mixed carbonaceous and
        ammonium sulfate particles. A better solution was achieved using the combined PMF analysis; the errors of the OA PMF
        solution were on average 12 % and they considerably improved for the combined PMF analysis (8 %). Furthermore, better
        correlation between the factors obtained by the latter analysis and the external data from collocated instruments was
        achieved, while the primary factors retrieved from the two analyses were highly correlated (R–Pearson = 0.89 on average).
Overall, our study suggests that the combined matrix PMF analysis is an effective approach to gain more insights into the
        nature and origin of ambient aerosols while applying PMF over a long–term dataset of NRS. Integrating the methodology of





rolling window simulations on combined datasets over different spatial and temporal variables will enlighten our knowledge on ambient aerosols and will enable us to extract the best of information that can be obtained by real–time long–term measurements from instruments like real time mass spectrometers and aerosol chemical speciation monitors.

## Data Availability

Data are available upon request to the author (o.zografou@ipta.demokritos.gr).

## Competing Interest

The authors declare that they have no conflict of interest.

## Author Contribution

OZ performed the data curation, formal analysis, visualization and writing the original draft; ACK and KE helped finding acquisition; OZ, ACK and MG performed the investigation; GC and OZ provided the methodology; KE helped with the project administration and resources; MIM and GC helped in running SoFi Pro and helped with discussions on the results; OZ, MG, MIM, GC and ACK provided conceptualization, supervision and validation; MG, MIM, GC, ACK, KE, ED and AP contributed in reviewing and editing the manuscript.

## Acknowledgments

This work was supported by COLOSSAL CA16109 and Co-financed by the European Union and Greek national funds through the Operational Program Competitiveness, Entrepreneurship and Innovation, under the call RESEARCH – CREATE – INNOVATE (project code:T1EDK-03437).

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
