# Peer review of "Combined organic and inorganic source apportionment on yearlong ToF-ACSM dataset at a suburban station in Athens"

_Atmospheric Measurement Techniques, 2022_

## Author Comment (AC2)

**RC2**

This work presents a valuable yearlong ToFACSM dataset for organic and inorganic source apportionment in Athens. It provide valuable information for the source of organic and inorganic source in Athens. The manuscript is generally well written, and falls well within the scope of AMT. Thus, it is of value to be published. After said that, some modifications are necessary to clarify what values the work add.

We sincerely thank the reviewer for the helpful comments on our manuscript and for deeming it as well written and of value to be published. Below, we reply to each comment (blue font) and we cite the changed parts as appear in the revised manuscript (grey font).

**Major comments**

This manuscript exhibits valuable measurements for organic and inorganic source apportionment, while it lacks focus. It may be helpful to strengthen how this dataset is useful for environmental studies or climate studies. Besides, it would be better to provide some implications on what values this dataset adds compared to previous studies. In summary, the authors should provide more discussions to strengthen the values of the dataset. Besides just providing values, it is also valuable to clarify why these values are valuable.

The reviewer has focused on some points which can be extremely beneficial for the manuscript. We thank the reviewer for the recommendations. We refer here to the parts of the manuscript where the strength of this methodology is pointed out.

Line 27-30: "This work applies a new methodology to a year-long ACSM dataset, provides insights on the sources of the non–refractory species of ambient aerosol and using innovative tools for applying PMF (Rolling window) enables the study of the temporal variation of these sources and also the variability of their composition."

Line 114-116: "Nevertheless, a long period of combined organic and inorganic source apportionment study spanning over a period of a year has not yet been published, leaving a gap in the comprehensive understanding of ambient aerosol sources, formation processes and mixing."

Line 701-703: "The peaks of the previously resolved HOA (Sect. 4.2) that were attributed to poor separation of the OA factors were not present in the combined matrix analysis, highlighting the improvement of the solution."

Line 957-962: "As regards the significance of the combined PMF analysis over organics PMF, it was shown that incorporating the inorganics in the PMF analysis valuable information regarding the mixing of organics and inorganics over time and the sources of total non-refractory species of PM1 can be obtained, while at the same time maintaining the quality of the solution obtained."

Although, we understand the point of the reviewer and therefore a new section (4.4) was added that compared the two analyses and stated more the value added by this work:

Line 872-909: "Integrating the inorganics in the PMF analysis adds valuable information concerning the mixing characteristics of organic and inorganic species over time, while rendering results that are qualitatively comparable to the widely-used organic aerosol PMF. Obtaining a better understanding on the sources and evolution processes of the total NRS, instead of merely OA, by applying source apportionment methods in combined organic and inorganic datasets for various site locations and for long-term datasets can be proven beneficial for atmospheric studies and climate models. The two analyses applied in the present study provided acceptable solutions both in terms of uncertainty (spread) of the factors and in terms of residuals, as explained in the paragraphs below.

…

Moreover, the primary factors obtained by both analyses were highly correlated with each other in terms of temporal variation, suggesting that the inclusion of the inorganics in the PMF scheme did not adversely affect the quality of the initial solution. More specifically, the time series of the HOA factor obtained from the combined matrix presented good correlation with the organic matrix resolved HOA factor (R–Pearson = 0.87). Combined COA time series agreed with the previously resolved COA time series (R–Pearson = 0.92). The BBOAs resolved from the two different analyses were highly correlated with each other (R–Pearson = 0.88). Finally, the correlation between the factors obtained and external tracers appears in Table S6, from which a slight improvement can be seen for spring. For the other seasons the correlations are stable between the factors from the two analyses and their respective external tracers, which confirms the successful deconvolution of the primary factors by both analyses."

**Minor comments.**

**Abstract, The method has been proposed in previous studies, but you applied it in a yearlong investigation in Athens. Thus, "new methodology" is not proper.**

We thank the reviewer for the comment. Indeed, in this work we apply this new methodology, we do not present a new one. This sentence was rephrased as:

This work applies a new methodology to a year-long ACSM dataset.

**Line 41. Brown carbon is also organic aerosol, while it absorbs light from ultraviolet to visible region.**

We agree with the reviewer and the sentence was revised:

"For example, black carbon can absorb light at all wavelengths, brown carbon absorbs ultraviolet and visible radiation (Moosmüller et al., 2009), while organic aerosol (except for brown carbon), nitrate and sulfate particles are responsible mainly for light scattering (Cabada et al., 2004)."

**Line 46. "Organic fraction usually comprises the greatest fraction of ambient aerosol", is it correct for all the regions?**

We thank the reviewer for the comment. The sentence was altered as follows:

"The organic fraction comprises 20-90 % of ambient fine aerosols (Kanakidou et al., 2005, Chen et al., 2022)."

**Line 48 "Secondary" should be "Secondary organic aerosols".**

We agree with the reviewer that this phrase was incomplete and he revised it:

"Secondary organic aerosols are the organic aerosols that are…"

**Line 51 -52 "aerosol forming" should be "aerosol-forming".**

We thank the reviewer for the comment. The revision of the phrase follows:

"…which then condense onto pre-existing aerosol-forming secondary organic aerosols (SOAs)."

**Line 53. "SOAs are the dominant form of organic aerosols", is it really correct? Please provide the references.**

We appreciate the reviewer's comment. The sentence was removed, since it is not a globally valid point.

**Line 54 – 58, please provide references to verify your clarifications.**

We thank the reviewer for the comment and agree that references are important here to verify our clarifications. The paragraph was revised:

"Secondary sulfates are found in the atmosphere mainly in the forms of $(NH_4)_2SO_4$ and $NH_4HSO_4$, after the neutralization of sulfuric acid by ammonia (Biggins and Harrison., 1979). Ambient ammonium nitrate is formed through the oxidation of anthropogenic NOx emissions (NO and $NO_2$) to nitric acid ($HNO_3$), which eventually reacts with ammonia ($NH_3$) (Stelson et al., 1979)."

**Please check the English grammar. Some examples are shown in the following.**

We thank the reviewer for pointing out some grammatical infelicities of the manuscript.

**Line 58 "in agriculture etc" should be "in agriculture, etc"**

The reviewer is correct and the sentence was revised:

"…pesticides in agriculture, etc."

**Line 60 "are also released in" should be "are also released into".**

We thank the reviewer for the comment.

"Chloride containing particles are also released into the atmosphere…"

**Line 82 you said "Although source apportionment studies on organic aerosols for long periods have been prevailing in recent years covering a wide range of different sites, a long period of combined organic and inorganic source apportionment has not yet been published.", but in line 70 you also say "Previous studies on particulate matter source apportionment in Greece have mainly focused on inorganic datasets". I know you mean that the aims of this work is to combine organic and inorganic source apportionment for a yearlong investigation. However, the logic of the sentence should be re-arranged.**

We thank the reviewer for the comment. It needs to be clarified that the sentence "Previous studies on particulate matter source apportionment in Greece have mainly focused on inorganic datasets" refers to the classical SA approaches on sample collection analysis and not in inorganic species measured by high time resolution instruments like the ACSM. The species included in this case are elemental component of PM and possibly ionic component and carbonaceous species as total EC, OC.

**Line 82 – 84: This sentence is too long.**

The reviewer brings up a helpful comment for this sentence which was split into two sentences as follows:

"Long period source apportionment studies on organic aerosols in recent years have covered a wide range of sites. Nevertheless, a long period of combined organic and inorganic source apportionment study spanning over a period of a year has not yet been published has not yet been published, leaving a gap in the comprehensive understanding of ambient aerosol sources, formation processes and mixing."

**Line 85 – 86: It would be better to re-write this sentence as two sentences. "one on the combined" should be "another on the combined"**

We thank the reviewer for the helpful comment.

"This study is the first one to present the results of two PMF analyses, one on the organic fraction and another on the combined organic and inorganic dataset of a ToF-ACSM for one year. The technique of the rolling window was also enabled in order to examine the temporal variability and the varying composition of the combined factors."

**Line 93 "member of" should be "a member of".**

We thank the reviewer for the comment.

"… (DEM), a member of the..."

**Line 96 "North east" should be "Northeast".**

The reviewer is correct for pointing that out.

"…8 km to the Northeast of Athens city…"

**Line 107 "afterwards" should be "afterward".**

We thank the reviewer for the comment.

"…the data were afterward averaged…"

**Line 108 "principle of" should be "the principle of".**

We appreciate the reviewer's comment.

"…and the principle of operation is given…"

**Figure 3: the axis of different subfigures are too close.**

We thank the reviewer for this comment, the figure was replotted.

**Line 576: is "OA" "OOA"?**

We thank the reviewer for this comment, we meant to say oxidized oxygenated OA. The sentence was altered to:

"…one more oxidized (MO-OOA) and one less oxidized OOA…"

**Line 596 – 599: This sentence is too long. In addition, in the discussion, some specific discussions on how the dataset adds should be added.**

We thank the reviewer for the comment. The sentence was rearranges and the added value of this study is highlighted in the added last section (4.4):

"Integrating the inorganics in the PMF analysis adds valuable information concerning the mixing characteristics of organic and inorganic species over time, while rendering results that are qualitatively comparable to the widely-used organic aerosol PMF. Obtaining a better understanding on the sources and evolution processes of the total NRS, instead of merely OA, by applying source apportionment methods in combined organic and inorganic datasets for various site locations and for long-term datasets can be proven beneficial for atmospheric studies and climate models. The two analyses applied in the present study provided acceptable solutions both in terms of uncertainty (spread) of the factors and in terms of residuals, as explained in the paragraphs below."

---

## Author Response (AR1)

**RC1**

**This paper describes the analysis of a year long dataset acquired with a TOF-ACSM in Athens, Greece. The paper is well-written and the comparison of two different PMF approaches is interesting and worth publishing. There were a few sections that were confusing. Addressing the following comments should clear those up.**

We genuinely thank the reviewer for the insightful comments and for considering the manuscript well-written and worth publishing. We **present here our response to each comment (blue font) and we quote the respective part of the revised manuscript (grey font).**

**Specific comments:**

**Comment 1. Lines 119-123: This is not how CDCE is applied. It's not the ammonium to nitrate fraction that matters, but the fraction of ammonium nitrate to total mass loading. Fraction of ammonium nitrate is clearly below 0.4 (based on the time-series in Figure S1 and Table S2), but the particles are also not neutralized, so you need to consider the second part of CDCE for acidic aerosols. Based on Table S2, CE=0.5 is fine for winter and spring, but a bit low for summer (CE=0.55) and fall (CE=0.56). It will only be a 10% error or so to apply CE=0.5 for all seasons, but you should explain why you are doing that.**

The reviewer emphasizes an extremely important point that needed to be changed and clarified. We thank the reviewer for the comment.

The ANMF was calculated and not the ammonium to nitrate fraction as was incorrectly mentioned in the manuscript. This paragraph was revised as follows:

"The collection efficiency chosen depends on three parameters; firstly, on the particulate water content. To account for it, a Nafion drier was placed in the inlet line. CE also depends on the ammonium nitrate fraction of the aerosol (ANMF), which was calculated to be higher than 0.4 for 99.9 % of the data indicating that a constant CE value of 0.5 should be optimum. Finally it depends on the acidity of the aerosol. Based on that, the CE was calculated 0.52 for NDJF, 0.49 for MAM, 0.55 for JJA and 0.56 for SO, while for the yearlong period it was 0.52. Therefore, the constant value of 0.5 was selected and this small variability should not affect much the solution given that the overall uncertainty for CE is 30 % (Bahreini et al., 2009)."

**Comment 2. Lines 250-256: Interquartile to median is a measure of spread or variability, not error so I would not call it an error. Isn't the variability in the mass concentration a reflection of atmospheric variability? It's not clear to me that lower variability necessarily means the PMF factors are a better representation of atmospheric sources. It also seems like a disconnect with the previous paragraph where the combined matrix had higher residuals (worse PMF) but lower spread (better PMF?).**

We really thank the reviewer for this comment. The reviewer is right pointing out that this should not be called an error, thus we name this value of the interquartile to median as uncertainties of the modelled factor (Canonaco et al., 2021, Chen et al., 2021). Since me-2 does not provide one unique solution (and thus there is no true answer),

we always need to consider how many possible solutions there are within the acceptable limits and how variant they are from each other. The variability in this part refers to the variability of the many repeats of the model, which in turn can be translated as the uncertainty of the solution.

Additionally, variability arises from the many repeats for each data point due to the technique of random a-values and the bootstrap resampling strategy and also from the different solutions in different rolling windows.

Moreover, larger uncertainty, or more spread, is not necessarily indicative of worse PMF, but since all solutions within the spread are mathematically equivalent, higher spread indicates the variation of possible solutions. Since we do not know the true answer this can be translated to uncertainty. On the other hand, higher residuals by definition means worse modelling results.

This paragraph that was moved to the last section (4.4) was changed as follows:

Line 886: "Since PMF provides a range of possible solutions, there is the need to determine how many of these solutions are within the acceptable limits and how much they vary from each other. The variability in this part refers to the variability of the many repeats of the model that can be translated as uncertainty. Moreover, uncertainty is created by the generation of each time point many times after the application of the random a-values constraints, the resampling technique of bootstrapping and the technique of the rolling window. Thus, the ratio of the interquartile to the median concentration is used as a measure of this uncertainty (Canonaco et al., 2021)."

**Comment 3. Lines 436-451. This paragraph is very confusing. What is a subtracted f44-f43 plot? You are plotting f44 vs f43 for OOA factors so why do you need to subtract HOA? Or do you mean you subtracted an HOA factor for the raw data points in the background of the Figure S8? In Figure S8, the markers are color coded by month, not concentration. There is a shading scale for concentration in the legend, but it is impossible to see in the figure. Maybe use size for concentration? The triangle plot does not evaluate the goodness of the PMF fit. It shows if the PMF profiles obtained here are consistent with typical ambient measurements. The description of the trends as vertical or horizontal is not very helpful. Please describe in terms of chemistry, i.e., as more or less oxygenated.**

The authors thank the reviewer for the comment. This kind of plots is widely used in relevant studies (Canonaco et al., 2015, Chen et al., 2021) for understanding how OOA factors adapt to the variations of $f44$ vs $f43$ using rolling PMF. The equation that this is based upon, as also found in the supplement of Chen et al (2021), is the following:

$$Subtracted\ f44 = \frac{mass\ concentration\ of\ OOA\ at\ \frac{m}{z}44}{mass\ concentration\ of\ OOA\ at\ \frac{m}{z}44 + residual\ of\ total\ OOA}$$

$$Subtracted\ f43 = \frac{mass\ concentration\ of\ OOA\ at\ \frac{m}{z}43}{mass\ concentration\ of\ OOA\ at\ \frac{m}{z}43 + residual\ of\ total\ OOA}$$

The reason for the subtraction of the primary factors is that the fractions at $m/z$ 44 and $m/z$ 43 are usually the dominating ions in OOA factors.

Nevertheless, the authors decided that this plot was more confusing than adding value to the description of the dataset and fell outside of the scope of this study and therefore it was removed.

**Specific minor comments:**

**Line 30: This methodology has been published before, so rather than saying this "work presents a new methodology," say "this work applies a new methodology to a year-long dataset."**

The authors thank the reviewer for the comment. This sentence was rephrased as:

This work applies a new methodology to a year-long ACSM dataset.

**Line 49-50: Do you mean IVOCs? VOCs are typically emitted already in the gas-phase.**

We thank the reviewer for this comment. This part of the sentence was deleted, since the authors agree that VOCs are already in the gas phase.

**Figure 1: I don't find pictures of instrument containers particularly useful and the contrast in the upper left panel is poor. I would suggest making this a 2-panel figure, one panel with a map that shows a scale between the upper left and upper right panels and one panel with the lower right photo. You could put the prevailing winds on the map, at least for winter and spring. In the lower right photo, label the yellow dot and indicate which direction is north.**

The authors thank the reviewer for the suggestion and agree that this adaptions will be helpful. This figure was replaced with a 2-panel and a compass was added to indicate north. The prevailing winds for each season appear in the supplement (Fig. S3) and the authors think it would not fit to insert the winds in this figure:

[Figure]

**Line 107: What was the detection limit for your instrument? Which species is this detection limit for?**

We thank the reviewer for the comment. The detection limit for each species was added in a sentence:

The 10 min detection limit for each species measured with the ToF-ACSM is 0.062 for organics, 0.006 for $SO_4^{2-}$, 0.007 for $NO_3^-$, 0.058 for $NH_4^+$ and 0.003 for $Cl^-$ (Fröhlich et al., 2013).

**Line 127: Please describe the approach rather than requiring someone to read the Supplement. Maybe replace "as described in the Supplement (Sect. S1)." with "from the wavelength dependence of the absorption (Sect. S1)."**

The paragraph of Section S1 was moved to the manuscript to avoid referring the reader to the supplement to check the approach followed.

**Line 168-171: This is a very long sentence. Replace "technique application a technique that" with "technique that" and put a period after "size." Then start a new sentence with "Calculations are repeated…"**

We thank the reviewer for this comment. The two sentences were separated as follows:

"SoFi Pro includes the rolling window technique that allows the user to track the variability of the factors by applying a window with selected length (usually 7, 14 or 28 days, depending on the size of the studied dataset) that moves with a chosen step. Calculations are repeated in that moving span providing the temporal changes in both profile and time series of the factors (Canonaco, 2021)."

**Line 174: Since the previous sentence mentions both wind and air mass analysis, indicate which one you are applying CPF to. Maybe start this sentence with "The wind analysis used the" and delete "was used"**

The authors thank the reviewer for the comment. The sentence was reshaped as follows:

"The wind analysis used the conditional probability function (CPF) to provide…"

**Line 182: 1000 m seems really high for back trajectory analysis of ground level measurements. Can you explain why you chose this height?**

The reviewer is right to point this out. The reason why the height of 1000 m AGL was chosen for the HYSPLIT runs was the relatively high altitude of the station (270 m ASL). The uncertainty due to the ground effects would be much if a lower height was to be chosen. Moreover, the trajectories were plotted for three different starting heights (500, 1000 and 1500 m) and the results were resembling.

**Line 191: The terminology in PMF can be confusing and I would define the terms in this introductory paragraph. Maybe something like "In the following, profile refers to the mass spectrum of a given factor and variable refers to an individual m/z."**

We find this comment helpful. The following sentence was added in order to explain better the PMF terms:

"In the following, profile refers to the mass spectrum of a given factor and variable refers to an individual mass to charge ratio ($m/z$)."

**Lines 192-202: The sentence about the mass concentration should go with the next paragraph that describes the mass loading calculation.**

We thank the reviewer for this comment. The sentence was moved to the end of the following paragraph.

**Line 200: You downweight the errors for the variables, not the whole species.**

The reviewer is correct to point out this fact. This statement was phrased as:

"The error values for each inorganic variable were downweighted…"

**Line 212: I'm not sure what "for each matrix" is referring to here. Maybe end the sentence after "number of factors."**

We thank the reviewer for this comment. This sentence was ended after "number of factors":

"The first step for source apportionment was to perform PMF analysis on the winter months (November–February) in order to identify the number of factors."

**Line 243: Maybe "subjective" or "qualitative" is better than "user-dependent" as in "the interpretation of PMF results is qualitative"**

We agree with the reviewer's statement. The word user-dependent was changed to "subjective".

**Line 247: Aren't OOAs and oxidized aerosols the same thing?**

We thank the reviewer for this comment. As oxidized aerosols are referred here the MOA and LOA factors that contain both organic and inorganic species. The sentence was rephrased to be clearer:

"The points at which the scaled residuals exceeded these thresholds were associated with peaks in SOAs in the OA PMF analysis and in oxidized aerosols (MOA and LOA) in the combined PMF analysis…"

**Lines 273-276: A step is missing in this description. Hysplit does not track species, only air masses. Do you mean that the Hysplit back trajectories are colored by the sulfate concentration at the end point? Also, the red blob looks more diffuse in winter, spring and summer than fall so I'm not sure about your conclusion that fall sulfate is more regional.**

The reviewer is right to point out that Hysplit does not track species. Having said that, we have not used the back trajectories independently, but as part of a trajectory statistical model (TSM), that makes it possible to identify source locations. The following part was added in the text:

"The basis of PSCF is that if a source is located at (i, j), an air parcel back trajectory passing through that location indicates that material from the source can be collected and transported along the trajectory to the receptor site. The PSCF is calculated as:

$$PSCF = n_{ij}/m_{ij} \qquad\qquad (8)$$

Where nij is the number of times that the trajectories passed through the cell (i, j) and mij is the number of times that a source concentration was high when the trajectories passed through the cell (i, j). The criterion for determining mij is based on the distribution of the measured values (e.g., upper quartile)."

Moreover, concerning the concentration of sulphate, we claim that it is a combination of regional transport, local formation and meteorology and the higher concentrations in spring, summer and autumn are due to local photochemical activity and less precipitation compared to winter:

"Simultaneously, sulfate levels were the result of regional transport, photochemical activity and local meteorology. The regional character of sulfate for all seasons is indicated by the HYSPLIT back trajectories in Fig. S2. In winter, sulfate values are lower due to enhanced precipitation, although regional sulfate was being transported to the station, while in the other seasons regional transport combined with local photochemical activity and less precipitation results in higher sulfate values (Stavroulas et al., 2019, Theodosi et al., 2018; Cusack et al., 2012, Dayan et al., 2017)."

**Lines 294-296: Since the sulfate is not neutralized with ammonia that suggests more local than regional sources. It would be more accurate to say that the flat diurnal in winter is consistent with regional sources. The afternoon peak in spring suggests local photochemical activity. The increase at night in summer and fall is due to larger changes in boundary layer height compared to other seasons. Make sure that your conclusions from the diurnals are consistent with your conclusions from the Hysplit back trajectories.**

The authors thank the reviewer and agree with the suggestions on the explanation of the diurnal behavior of sulphate, thus it was changed to:

"...the diurnal profile of sulfate was flat in winter, consistent with regional sources and meteorological conditions that do not favor local photochemical activity. In spring, summer and early autumn sulfate presents a diurnal structure that is related to local photochemical activity and boundary layer height"

**Lines 297-299: The formation of NH4Cl would be higher at night when the temperature is lower. How does that explain an afternoon peak?**

We thank the reviewer for the comment. The term used was wrong, the second peak mentioned is an evening peak and not an afternoon peak since it is observed at 20:00. Thus it is due to NH$_4$Cl formation when the temperature drops. Therefore, the sentence was rephrased as follows:

"Chloride presented two distinct peaks (i.e. morning and evening) for all the seasons, related to temperature–dependent gas–particle partitioning of chlorine (i.e. chloride is primarily detected as ammonium chloride), biomass burning emissions and prevailing atmospheric conditions."

**Line 339: Change the section heading to make it clear that this section is about the OA PMF analysis, not organic aerosols in general.**

The authors thank the reviewer for this comment. The heading was changed to:

"PMF analysis of organic aerosols"

**Line 341: Be more specific, e.g., "the mass spectrum from m/z 100 to 200 is depicted"**

We thank the reviewer for this comment. This sentence was rephrased as:

190    "The profiles of the five factors for *m/z* 10 to 100 are presented…"

**Figure 3: It is difficult to see what is going on in the panel with both BBOA and temperature. I would suggest a separate y-axis for temperature. Or, alternatively, you could average the temperature to daily values since it is the seasonal trend that matters.**

We thank the reviewer and agree with this point, thus the figure was changed; the temperature wave was replaced
195    with the daily averaged temperature as shown below:

[Figure]

**Lines 377-382: Make it clear that these previous studies are high resolution analysis that gives specific ion fragments at m/z 55 and 57. Also, not clear why you are giving chemical names of gas-phase species for ions from particles. I would delete the chemical names.**

200    The authors thank the reviewer and highly agree with this point. Although, upon closer examination this sentence
was not considered as very useful and was removed from the manuscript by the authors.

**Line 387: Why surprising? It is the same site.**

We thank the reviewer and agree that the use of this word was incorrect and was therefore removed.

**Line 394: What does "with a total duration of 8 h" mean? Are you comparing only 8 hours of data? If so,**
205    **specify which 8 hours.**

We thank the reviewer for this comment. The loading is referred to the hours that cooking or traffic emissions are observed compared to the hours of the day when the concentration of COA and HOA drops. It means the duration of the loading effect of this emission.

**Line 452: Change the section heading to make it clear that this section is about the combined PMF analysis, not submicron aerosols in general.**

The authors thank the reviewer for this comment. The heading was changed to:

"PMF analysis of submicron aerosol"

**Figure 4. Would suggest averaging the temperature data to make the panel with both T and BBOA clearer. It's very hard to see RH and T in the panel with AN. I would make a separate panel. The caption should say "aerosol factors" not "sources."**

The authors thank the reviewer for this helpful suggestion. Figure 4b was changed after averaging temperature with daily resolution. Nevertheless, the part of the manuscript about the dependence of ammonium nitrate on temperature was removed, therefore the plot 4c was replaced by the diurnals of the factors without the interference of temperature and relative humidity:

[Figure]

[Figure]

The caption was changed to:

"Figure 4. Mass spectra (a), time series (b) and diurnal plots (c) of the seven submicron factors and (d) diurnal plot of AmNi with Relative Humidity and Air Temperature."

**Lines 472-474: Many of these changed by 0.01. Is that significant? Maybe point out one or two cases where there was a significant improvement.**

We thank the reviewer for the comment and agree that this changes were not all significant. This sentence was moved to the new section (4.4) and replaced by the following:

"Finally, the correlation between the factors obtained and external tracers appears in Table S6, from which a slight improvement can be seen for spring. For the other seasons the correlations are stable between the factors from the two analyses and their respective external tracers."

**Line 507: Insert "The CPF polar plot in" before "Figure S10" so the reader does not have to go to the SI to find out what S10 is.**

The authors thank the reviewer for the comment. The phrase "The CPF polar plot in" was added before Fig. 10:

"The CPF polar plot in Fig. S10 further confirms that ammonium nitrate was locally formed."

**Line 522: You need something to transition between these two sentences. Maybe "As noted above for the OA PMF analysis,"**

The authors thank the reviewer for the comment. The phrase "As noted above for OA PMF analysis" was added in the beginning of the sentence.

240  "As noted before for the OA PMF analysis, LO-OOA and MO-OOA, retrieved by PMF analysis on the organic fraction, were well correlated with…"

**Line 536: I think you mean the "attribution of organics" rather than the "contribution of organics." This sentence should end in a comma, not a period, because the next sentence is not a complete sentence.**

We thank the reviewer for this comment. The word "contribution" was replaced with the word "attribution" and
245  the following sentence was added:

"Overall, the attribution of organics to MOA and LOA followed the same trend as that of MO-OOA and LO-OOA, while the difference in absolute values between MO-OOA/MOA and LO-OOA/LOA concentrations was mainly driven by the sulfate apportioned to the combined secondary factors."

**Figure 5. Use the same y-axis scaling for both panels so that it is easier to interpret the discussion in the**
250  **text.**

The authors thank the reviewer for this suggestion. The same y-axis was used for the two panels and the figure was replaced:

[Figure]

**Lines 557-568: This whole paragraph seems like it should go earlier in this section since it is more of an**
255  **overview of the results. Also, some of it repeats information that you have already discussed in more detail.**

**And the phrase "will be discussed below" doesn't make any sense since this is the last paragraph of the results.**

The authors thank the reviewer for this comment and highly agree that the repositioning of this paragraph will benefit the manuscript. Moreover, some parts that were already discussed in previous sections needed to be
260 deleted. Thus, the first part of this paragraph was moved at the beginning of the Section 4.3. The rest of it, that referred to Fig. S10 was deleted, because it was also present in the factors description above and the reader was referred to Fig. S10 when it was needed in the above paragraphs of the Section 4.3:

Lines 701: …small contribution of inorganic species in this traffic–related factor (Fig. S10).

Line 706: As also shown in Fig.S10, this factor consisted mainly of organics…

265 Line 712: In this factor the inorganics presented lower contribution than on the other two POA factors (Fig. S10).

Line 717: The ammonium nitrate factor resolved in this study was composed of 55 % $NO_3^-$ and 18 % $NH_4^+$ (Fig. S10)

Line 804: A factor predominantly composed of sulfate and ammonium was retrieved in this study. 64 % of the mass of this factor was attributed to $SO_4^{2-}$ and 19.5 % $NH_4^+$ (Fig. S10).

270 The last part discussing Fig. S11 was left as the last paragraph of the section 4.3.

**Table S1: Why is SFBOA in this table? It is not discussed in the paper.**

The authors thank the reviewer for pointing this out. SFBOA was a typing mistake and it was replaced by BBOA.

**Figure S1: It would be helpful to indicate the seasons with vertical bars.**

We thank the reviewer and agree that it would be helpful to revise this plot in the way proposed. The seasons
275 were added as shaded vertical bars in this figure:

[Figure]

**Figure S2. I would expand the maps so the detail is clearer. Asia and Africa are not relevant to the discussion. Also, it would be nice to have the season label on each panel.**

We thank the reviewer and agree with this suggestion. The maps were expanded to show more clearly the scale that is relevant for the discussion and also seasons were labeled on each panel:

[Figure]

**Figure S4. It would help to have species labels on each panel. Same comment for Figure S7.**

The authors thank the reviewer for this comment. Species labels were added in Figures S4 and S7 as recommended:

[Figure]

**Table S4: Since NH4 is a linear combination of SO4 and NO3, and in this data dominated by SO4, correlations with NH4 do not add any new information. I would delete the two NH4 rows.**

We thank the reviewer for the comment. The NH4 rows of the table were deleted, because as stated by the reviewer and agreed by the authors they didn't add any new information.

**Figure S8: Explain what the black dots are – is this the subtracted data? The shading for size is not visible. Maybe use marker size instead? Not clear what "on the upper size," "on the lower size," and "in the middle" mean. I would delete.**

The authors thank the reviewer for the comment. The black dots were the subtracted data, but as discussed before it was decided that this plot be removed.

**Table S5. Make sure that the tables are in the same order as mentioned in the text.**

**We thank the reviewer for this helpful suggestion. The order that the tables are mentioned in the text was corrected.**

**Technical corrections:**

**We deeply thank the reviewer for the thorough revision of the manuscript and the truly helpful technical revisions proposed below.**

**All the technical corrections were made following the reviewer's comments:**

305     **Line 28: No "s" at the end of "aerosols"**

      **Line 37: "causes of" is better than "reasons for"**

      **Line 44: No "s" at the end of "clouds"**

      **Line 45: Start the sentence with "The"**

      **Line 58: No "s" at the end of "vehicles"**

310     **Line 64: Start the sentence with "The"**

      **Lines 64-67: This sentence would be easier to understand if you delete "by real-time measurements" at the end and insert "real-time" before "quantification"**

      **Line 66, 68, 93: Define all acronyms: PM1, PMF, GAW, ACTRIS, PANACEA**

      **Line 78: "fingerprint" not "footprint"**

315     **Line 96: replace "North east from" with "northeast of"**

      **Line 113: Particles are focused into a "narrow beam" not a "narrow airbeam"**

      **Line 115: Delete "in a tungsten filament" The electrons come from a tungsten filament, but the ionization doesn't occur in the filament.**

      **Line 115: Replace "following" with "according to"**

320     **Line 118: Should be a comma before "while" rather than a period before "While"**

      **Lines 146-149: I would put Eq 1 after the sentence that ends "G and F (Eq.1)."**

      **Line 201: Add "s" to "variable"**

      **Line 211: Add "the" before "winter"**

      **Line 214: Delete "s" at the end of "factors"**

325     **Line 226: Do you mean "lunchtime" or "mealtime"? It's not "noon" if it's 14:00.**

      **Line 228: Insert "of" between "fractions" and "m/z"**

      **Line 246: Replace "was well-fitted" with "fit the data well"**

**Line 265:** Replace "at" with "in"

**Line 278:** Replace "portioning" with "partitioning"

**Lines 285-287:** The end of this sentence is confusing. Do you mean "except for sulfate, for which concentrations were more similar."

**Line 287:** Replace "rest" with "other"

**Lines 292-293:** This is a run on sentence. Replace "night, however" with "night. However,"

**Line 352:** Not clear what "distinctly recognized" means. Do you mean "identified based on its distinctive mass spectrum"

**Line 355:** Delete the chemical formulas since you have them in parentheses later in the sentence.

**Line 357:** Replace "most" with "quite"

**Line 360:** Insert "with the" after "correlated"

**Line 363:** Insert "The" at the beginning of the sentence.

**Line 365:** Insert "the" before "HOA factor"

**Line 385:** Replace "in" with "at"

**Line 392:** Replace "configuring right" with "separating"

**Line 394:** Replace "respective" with "loading"

**Line 406:** Replace "to" with "with"

**Line 412:** Replace "were very relevant to" with "was highly correlated with"

**Line 416:** Replace "at" with "in"

**Line 417:** Insert "the" before "BBOA"

**Line 424:** Insert "The" before "MO-OOA"

**Line 427:** Do you mean Figure 3c?

**Line 428:** Replace "on" with "to"

**Lines 455-456: Replace "daily trends…concentration" with "diurnal trends (Fig. 4c) of each factor"**

**Line 471: Replace "resembles significantly to" with "is similar to"**

**Line 472: Replace "configuration of this factor with" with "agreement of this factor between"**

**Line 476: Add "s" to "contribution"**

**Line 478: Delete "highly"  before "agreed"and "'s" on "COA"**

**Line 502: "the ammonium nitrate peak" sounds better than "ammonium nitrate's peak"**

**Line 505: Replace "leaded" with "led"**

**Line 506: Delete "'s" on "nitrate"**

**Line 510: "predominantly" not "dominantly"**

**Lines 530-531: Delete "highly" Delete "s" on "SOAs" Delete "particles" after "sulfate"**

**Line 534: "and higher" might be better than "while peaked"**

**Line 573: Replace "sources" with "factors"**

**Line 577: Open parenthesis missing.**

**Line 586: Replace "simulations" with "sources"**

**Line 591: Replace "errors" with "variability" or "spread"**

**Line 598: Replace "on ambient aerosols…best of information that can be obtained by" with "of ambient…best information from"**

**Line 605: Replace "finding" with "funding"**

Line 957-962: "As regards the significance of the combined PMF analysis over organics PMF, it was shown that incorporating the inorganics in the PMF analysis valuable information regarding the mixing of organics and inorganics over time and the sources of total non-refractory species of PM1 can be obtained, while at the same

440 time maintaining the quality of the solution obtained."

Although, we understand the point of the reviewer and therefore a new section (4.4) was added that compared the two analyses and stated more the value added by this work:

Line 872-909: "Integrating the inorganics in the PMF analysis adds valuable information concerning the mixing characteristics of organic and inorganic species over time, while rendering results that are qualitatively comparable to the widely-used organic aerosol PMF. Obtaining a better understanding on the sources and evolution processes of the total NRS, instead of merely OA, by applying source apportionment methods in combined organic and inorganic datasets for various site locations and for long-term datasets can be proven beneficial for atmospheric studies and climate models. The two analyses applied in the present study provided acceptable solutions both in terms of uncertainty (spread) of the factors and in terms of residuals, as explained in the paragraphs below.

…

Moreover, the primary factors obtained by both analyses were highly correlated with each other in terms of temporal variation, suggesting that the inclusion of the inorganics in the PMF scheme did not adversely affect the quality of the initial solution. More specifically, the time series of the HOA factor obtained from the combined matrix presented good correlation with the organic matrix resolved HOA factor (R–Pearson = 0.87). Combined COA time series agreed with the previously resolved COA time series (R–Pearson = 0.92). The BBOAs resolved from the two different analyses were highly correlated with each other (R–Pearson = 0.88). Finally, the correlation between the factors obtained and external tracers appears in Table S6, from which a slight improvement can be seen for spring. For the other seasons the correlations are stable between the factors from the two analyses and their respective external tracers, which confirms the successful deconvolution of the primary factors by both analyses."

**Minor comments.**

**Abstract, The method has been proposed in previous studies, but you applied it in a yearlong investigation in Athens. Thus, "new methodology" is not proper.**

We thank the reviewer for the comment. Indeed, in this work we apply this new methodology, we do not present a new one. This sentence was rephrased as:

This work applies a new methodology to a year-long ACSM dataset.

**Line 41. Brown carbon is also organic aerosol, while it absorbs light from ultraviolet to visible region.**

We agree with the reviewer and the sentence was revised:

"For example, black carbon can absorb light at all wavelengths, brown carbon absorbs ultraviolet and visible radiation (Moosmüller et al., 2009), while organic aerosol (except for brown carbon), nitrate and sulfate particles are responsible mainly for light scattering (Cabada et al., 2004)."

**Line 46. "Organic fraction usually comprises the greatest fraction of ambient aerosol", is it correct for all the regions?**

475 We thank the reviewer for the comment. The sentence was altered as follows:

"The organic fraction comprises 20-90 % of ambient fine aerosols (Kanakidou et al., 2005, Chen et al., 2022)."

**Line 48 "Secondary" should be "Secondary organic aerosols".**

We agree with the reviewer that this phrase was incomplete and he revised it:

"Secondary organic aerosols are the organic aerosols that are…"

480 **Line 51 -52 "aerosol forming" should be "aerosol-forming".**

We thank the reviewer for the comment. The revision of the phrase follows:

"…which then condense onto pre-existing aerosol-forming secondary organic aerosols (SOAs)."

**Line 53. "SOAs are the dominant form of organic aerosols", is it really correct? Please provide the references.**

485 We appreciate the reviewer's comment. The sentence was removed, since it is not a globally valid point.

**Line 54 – 58, please provide references to verify your clarifications.**

We thank the reviewer for the comment and agree that references are important here to verify our clarifications. The paragraph was revised:

"Secondary sulfates are found in the atmosphere mainly in the forms of $(NH_4)_2SO_4$ and $NH_4HSO_4$, after the
490 neutralization of sulfuric acid by ammonia (Biggins and Harrison., 1979). Ambient ammonium nitrate is formed
through the oxidation of anthropogenic NOx emissions (NO and $NO_2$) to nitric acid ($HNO_3$), which eventually
reacts with ammonia ($NH_3$) (Stelson et al., 1979)."

**Please check the English grammar. Some examples are shown in the following.**

We thank the reviewer for pointing out some grammatical infelicities of the manuscript.

495 **Line 58 "in agriculture etc" should be "in agriculture, etc"**

The reviewer is correct and the sentence was revised:

"…pesticides in agriculture, etc."

**Line 60 "are also released in" should be "are also released into".**

We thank the reviewer for the comment.

500 "Chloride containing particles are also released into the atmosphere…"

**Line 82 you said "Although source apportionment studies on organic aerosols for long periods have been prevailing in recent years covering a wide range of different sites, a long period of combined organic and inorganic source apportionment has not yet been published.", but in line 70 you also say "Previous studies on particulate matter source apportionment in Greece have mainly focused on inorganic datasets". I know**
505 **you mean that the aims of this work is to combine organic and inorganic source apportionment for a yearlong investigation. However, the logic of the sentence should be re-arranged.**

We thank the reviewer for the comment. It needs to be clarified that the sentence "Previous studies on particulate matter source apportionment in Greece have mainly focused on inorganic datasets" refers to the classical SA approaches on sample collection analysis and not in inorganic species measured by high time resolution
510 instruments like the ACSM. The species included in this case are elemental component of PM and possibly ionic component and carbonaceous species as total EC, OC.

**Line 82 – 84: This sentence is too long.**

The reviewer brings up a helpful comment for this sentence which was split into two sentences as follows:

"Long period source apportionment studies on organic aerosols in recent years have covered a wide range of
515 sites. Nevertheless, a long period of combined organic and inorganic source apportionment study spanning over a period of a year has not yet been published has not yet been published, leaving a gap in the comprehensive understanding of ambient aerosol sources, formation processes and mixing."

**Line 85 – 86: It would be better to re-write this sentence as two sentences. "one on the combined" should be "another on the combined"**

520 We thank the reviewer for the helpful comment.

"This study is the first one to present the results of two PMF analyses, one on the organic fraction and another on the combined organic and inorganic dataset of a ToF-ACSM for one year. The technique of the rolling window was also enabled in order to examine the temporal variability and the varying composition of the combined factors."

525 **Line 93 "member of" should be "a member of".**

We thank the reviewer for the comment.

"… (DEM), a member of the..."

**Line 96 "North east" should be "Northeast".**

The reviewer is correct for pointing that out.

530 "…8 km to the Northeast of Athens city…"

**Line 107 "afterwards" should be "afterward".**

We thank the reviewer for the comment.

"…the data were afterward averaged…"

**Line 108 "principle of" should be "the principle of".**

535 We appreciate the reviewer's comment.

"…and the principle of operation is given…"

**Figure 3: the axis of different subfigures are too close.**

We thank the reviewer for this comment, the figure was replotted.

**Line 576: is "OA" "OOA"?**

540 We thank the reviewer for this comment, we meant to say oxidized oxygenated OA. The sentence was altered to:

"…one more oxidized (MO-OOA) and one less oxidized OOA…"

**Line 596 – 599: This sentence is too long. In addition, in the discussion, some specific discussions on how the dataset adds should be added.**

We thank the reviewer for the comment. The sentence was rearranges and the added value of this study is
545 highlighted in the added last section (4.4):

[revised manuscript text omitted]